# Improving estimation of a record breaking East Asian dust storm emission with lagged aerosol Ångström Exponent observations

Yueming Cheng[1,2,3], Tie Dai[1,2*], Junji Cao[1], Daisuke Goto[4], Jianbing Jin[3], Teruyuki Nakajima[5], Guangyu Shi[1]

[1]State Key Laboratory of Numerical Modeling for Atmospheric Sciences and Geophysical Fluid Dynamics, Institute of Atmospheric Physics, Chinese Academy of Sciences, Beijing, China
[2]Collaborative Innovation Center on Forecast and Evaluation of Meteorological Disasters, Nanjing University of Information Science and Technology, Nanjing, China
[3]Collaborative Innovation Center of Atmospheric Environment and Equipment Technology, Jiangsu Key Laboratory of Atmospheric Environment Monitoring and Pollution Control (AEMPC), Nanjing University of Information Science and Technology, Nanjing, China
[4]National Institute for Environmental Studies, Tsukuba, Japan
[5]Tokyo University of Marine Science and Technology, Tokyo, Japan
*Correspondence to:* Tie Dai (daitie@mail.iap.ac.cn)

**Abstract.** A record-breaking East Asian dust storm over recent years occurred in March 2021. Ångström Exponent (AE) which, measures the wavelength-dependence of aerosol optical thickness (AOT), is significantly sensitive to large aerosol such as dust. Due to the lack of observations during dust storms and the accuracy of satellite-retrieved AE depending on the instrument and retrieval algorithm, it is possible to estimate the dust storm emission using the time-lagged ground-based AE observations. In this study, the hourly AEs observed by Aerosol Robotic Network (AERONET) are assimilated with the ensemble Kalman smoother (EnKS) and Weather Research and Forecasting model coupled with Chemistry (WRF-Chem) to optimize the simulated dust emission from March 14 to 23, 2021. The results demonstrate that the additional inclusion of AE can optimize the size distribution of dust emission and the associated total flux, depending on the covariance between time-lagged AE observations and the simulated dust emission in each size bin. Validation by independent observations from Skynet Observation NETwork (SONET) shows that assimilating additional AE information reduces the root mean square error (RMSE) of simulated AOT and AE by approximately 17% and 61%, respectively, as shown by the comparison between the AOT DA-SZD and AOT+AE DA-SZD experiments. The temporal variation of both simulated AOT and AE are improved through assimilating additional AE information. The assimilation of AOT and AE also makes the magnitude and variations of aerosol vertical extinctions more comparable to the independent Cloud-Aerosol Lidar with Orthogonal Polarization (CALIOP) observations both in the westward and eastward pathways of dust transport. The optimized dust emission in the Gobi desert during this period is estimated as 52.63 Tg and reaches a peak value of 3837 kt h$^{-1}$ at 07:00 UTC on March 14.

## 1 Introduction

Mineral dust is the most abundant atmospheric aerosol component in terms of aerosol dry mass. It affects the climate system by scattering and absorbing longwave and shortwave radiation and also contributes to the formation of cloud condensation nuclei (CCN) and ice-nucleating particles (INP) (Huang et al., 2006; IPCC AR6, Kok et al., 2018; Liu et al., 2020). Dust also carries organic matter and transports iron to the ocean, which is vital to ocean productivity and ocean-atmosphere $CO_2$ exchange, thus inducing impacts on the cycles of dust and carbon (Shao et al., 2011). Dust deposition on snow surfaces can influence the snow albedo and modify the water cycle and energy budget (Wu et al., 2018; Kang et al., 2019; Wang et al., 2020). Moreover, severe dust storms can induce air pollution and affect human health (Chen et al., 2020).

East Asia, including the Taklimakan and Gobi deserts, is the world's second largest dust source, accounting for approximately 40% of the global dust emissions (Satake et al., 2004; Kok et al., 2021) and 88% of the dust in China and neighbouring seas (Han et al., 2022). Although the dust activities in East Asia have declined recently (Wu et al., 2022), the unexpected extreme dust storm event that occurred from March 14 to 23, 2021 has raised widespread concern. Numerical models are important tools for studying severe dust storms, and the dust emission is a significant parameter for characterizing dust activity. However, due to differences in the parameterizations of dust source fluxes, dust particle sizes, and model resolutions, simulated East Asian dust emissions vary by more than an order of magnitude among different models (Textor et al., 2006; Uno et al., 2006; Gliß et al., 2021; Kok et al., 2021), indicating that dust emission is a highly uncertain process in dust simulation. Data assimilation, which incorporates observation information into numerical models, provides a top-down method for optimizing estimates of dust emissions. Yumimoto et al. (2008) assimilated the dust extinction coefficients derived from ground-based lidar networks using a four-dimensional variational (4D-Var) method, increasing East Asian dust emissions by approximately 10 times. Sekiyama et al. (2010) developed an Ensemble Kalman Filter (EnKF) data assimilation system to jointly correct the global dust emissions and aerosol mixing ratios by assimilating vertical observations from the Cloud-Aerosol Lidar and Infrared Pathfinder Satellite Observations (CALIPSO). Wang et al. (2012) constrained the amount and location of dust emissions in the Taklimakan and Gobi deserts using the GEOS-Chem adjoint model by assimilating aerosol optical thicknesses (AOTs) from the Moderate Resolution Imaging Spectroradiometer (MODIS). Schutgens et al. (2012) assimilated observations from AErosol RObotic NETwork (AERONET) and MODIS to estimate emissions for dust, sea salt, and carbonaceous aerosols using an ensemble Kalman smoother technique. Yumimoto and Takemura (2015) performed inverse modeling of Asian dust with four-dimensional variational (4D-Var) data assimilation system and MODIS-retrieved AOT over ocean. Escribano et al. (2016) estimated dust emissions and reduced their uncertainty over the Sahara desert and the Arabian Peninsula by assimilating MODIS AOT retrievals. Di Tomaso et al. (2017) used the four-dimensional local ensemble transform Kalman filter (LETKF) to assimilate MODIS Dark Target and Deep Blue AOTs for improving dust analyses and forecasts on a global scale. More recently, high-frequency AOTs from the Himawari-8 geostationary satellite have been used for the optimization of dust storm emissions with a reduced tangent linearization 4D-Var technique (Jin et al., 2019).

In addition to AOT, Ångström Exponent (AE), which measures the wavelength-dependence of AOT and is sensitive to size of aerosol particles, may have a positive impact on data assimilation (Tsikerdekis et al., 2022, 2023). The estimated emission may be misrepresented by not including observations related to size (Chen et al., 2018, 2019; Tsikerdekis et al., 2021). However, most of the previous studies have focused on assimilating AOT to estimate new dust emissions, while few studies have explored the potential benefits of incorporating aerosol size information such as AE observations.

Therefore, how will the assimilation of the AE observations affects the optimization of the dust emission? It becomes an important scientific question. Due to the accuracy of satellite-retrieved AE depends on the instrument and retrieval algorithm, the ground-based AE is better than satellite-based AE and can be more useful for optimizing dust emissions.

Therefore, the AOT and AE observations from the ground-based AErosol RObotic NETwork (AERONET) are assimilated to investigate the sensitivity of dust emission to observed size information in this study. The additional benefit of aerosol size information in estimating dust emission is explored by only assimilating AOT and simultaneously assimilating AOT and AE. The experiments are performed using the Ensemble Kalman smoother (EnKS) assimilation framework (Dai et al., 2019) to constrain the extreme dust storm emission over East Asia from March 14 to 23, 2021. The Sun-Sky Radiometer Observation Network (SONET) and CALIPSO observations are used for independent validation.

Section 2 describes the assimilated and independent validation observations. Our dust emission optimization system and experimental design are presented in Sect. 3. Section 4 presents the optimized emission results and the validations using multi-sensor independent observations. The conclusions are given in Sect. 5.

## 2 Observation Data

### 2.1 Assimilated AERONET observations

AErosol RObotic NETwork (AERONET) is a federation of ground-based remote sensing aerosol network that collects aerosol optical observations with sun photometers from various stations globally (https://aeronet.gsfc.nasa.gov) (Holben et al., 1998; Giles et al., 2019). Version 3 AOT data are divided into three levels according to the data quality procedures: Level-1 (unscreened), Level-1.5 (cloud-screened and quality-controlled), and Level-2 (quality-assured). In this study, the Version 3 Level-2 AOT at 550 nm and AE at 440-870 nm from AERONET are assimilated. AOT ($\tau$) at 550 nm is obtained by logarithmic interpolation of the AOTs at 440 nm and 675 nm. AE ($\alpha$) at 440-870 nm is calculated with the AOT at 440 nm and 870 nm using the following equation: $\alpha_{440-870} = -ln(\tau_{870}/\tau_{440})/ln(870/440)$. To ensure accuracy, AE value is considered valid only when the AOTs at 440 nm and 870 nm both exceed 0.05 (Giles et al., 2019). The instantaneous observations are averaged every 1 hour, centering on the assimilation time slot. The observation error ($\epsilon$) attributed to this averaged observation is calculated by a representation error ($\epsilon_r$) and an instrument error ($\epsilon_o$) as $\epsilon^2 = \epsilon_r^2 + \epsilon_o^2$. Due to the representation error is related to the WRF-Chem grid resolution, the representation error in AERONET AOT and AE is calculated depending on the AOT and AE temporal variability of AERONET and WRF-Chem with 45 km horizontal resolution (Schutgens et al., 2010). By averaging results at all AERONET sites in March 2021, the relative AOT temporal

variations of AERONET and WRF-Chem in 1 h interval are $0.11\tau$ and $0.1\tau$, while the AE temporal variations of AERONET and WRF-Chem in 1 h interval are 0.05 and 0.02, respectively. Therefore, the representation errors in AERONET AOT ($\tau$)

100    and AE in the 1 h interval are $\epsilon_{AOTr} = 0.01\tau$ and $\epsilon_{AEr} = 0.03$, respectively. The instrument error in AOT ($\epsilon_{AOTi}$) is defined as 0.015 and the instrument error of AE ($\epsilon_{AEi}$) is estimated by propagating the instrument error in AOT at 440 and 870 nm as $\epsilon_{AEi} = \sqrt{((\epsilon_{AOTi}/\tau_{870})^2 + (\epsilon_{AOTi}/\tau_{440})^2)/(ln(870/440)^2)}$ (Schutgens et al., 2010).

To minimize the influences of anthropogenic emissions, only the AERONET AOT and AE dominated by dust are assimilated to optimize the dust emissions, which are chosen with AE at 440-870 nm less than 0.4 (Huneeus et al., 2011). In

105    addition, due to the uncertainties of modelled covariance between dust emission and aerosol optical properties increasing with the distance from the source region, only the observations within 2190 km (3.65 times of the localization length) of the East Asian dust source region are used for data assimilation. As shown in Fig. 1(a), there are 5 AERONET sites with available observations from 14 March to 23 March 2021 for data assimilation, including 4 sites named as Beijing-CAMS (39.93°N, 116.32°E), Beijing (39.98°N, 116.38°E), Beijing_PKU (39.99°N, 116.31°E), and Beijing_RADI (40.00°N,

110    116.38°E) in the downwind area and a site near the dust source region named as Dalanzadgad (43.58°N, 104.42°E). The assimilated AOT and AE values at the AERONET sites are also given in Fig. 1(b,d). For the Dalanzadgad site, the AOT values from 14 March to 17 March 2021 are significantly higher than those from 18 March to 23 March, while the AE values show the opposite features.

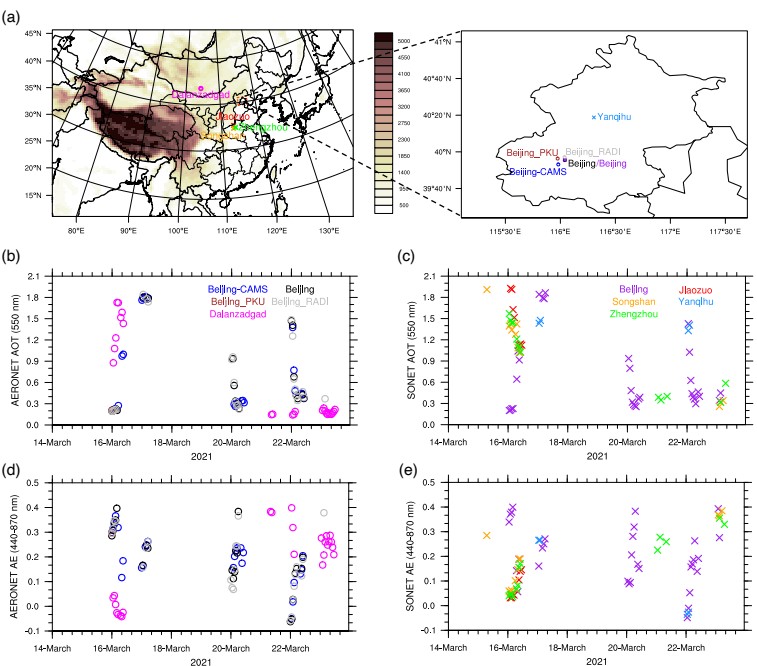

115    **Figure 1. Locations of selected Aerosol Robotic Network (AERONET) sites for data assimilation and Skynet Observation NETwork (SONET) sites for independent validation in this study (a). The color-filled contour represents the altitude (unit: m). AERONET sites are illustrated by circle and SONET sites are illustrated by cross. The hourly values of aerosol optical thickness**

(AOTs) at 550 nm and Angstrom Exponent (AEs) at 440-870 nm at AERONET (b,d) and SONET sites (c,e) during 14-23 March 2021.

## 2.2 Independent SONET and CALIOP observations

Sun-Sky Radiometer Observation Network (SONET) is a ground-based aerosol network employing Cimel radiometer and multiwavelength polarization measurement to provide long-term columnar atmospheric aerosol properties over China (www.sonet.ac.cn) (Li et al., 2018). The aerosol optical related products including AOT, AE, fine-mode fraction (FMF) are graded into three levels: Level-1 (no triplet), Level-1.5 (cloudy), and Level-2 (no cloud). In this study, the Level-2 AOT at 550 nm and AE at 440-870 nm from SONET are used for independent validation and the instantaneous observations are also averaged every hour to generate hourly AOT and AE datasets. As shown in Fig. 1(a), there are 5 SONET sites with available observations from 14 March to 23 March 2021, including: Yanqihu (40.40°N, 116.67°E), Beijing (40.00°N, 116.37°E), Jiaozuo (35.18°N, 113.20°E), Songshan (34.53°N, 113.09°E), and Zhengzhou (34.70°N, 113.66°E). The AOT and AE values at the SONET sites are also given in Fig. 1 (c,e). Similar to the Dalanzadgad site, the Jiaozuo, Songshan, and Zhengzhou sites experience a stronger dust process from 14 March to 17 March 2021 with higher AOTs and lower AEs.

Cloud-Aerosol Lidar with Orthogonal Polarization (CALIOP), aboard the CALIPSO satellite, is a dual-wavelength polarization lidar that performs observations of vertical structures of aerosols and clouds on a global scale (http://www-calipso.larc. nasa.gov) (Winker et al., 2007). In this study, the aerosol extinction coefficients at 532 nm in the CALIPSO lidar Level-2 Version 4.20 aerosol profile products over the altitude range below 12 km are also used for evaluation. The CALIPSO lidar Level-2 Version 4.20 Vertical Feature Mask (VFM) products, which include the feature types and subtype information, are used for aerosol discrimination (Omar et al., 2009; Cheng et al., 2019).

## 3 Model and data assimilation methodology

### 3.1 Forward model

The Weather Research and Forecasting model coupled with Chemistry (WRF-Chem) version 4.4 is served as the forward model. The model domain covers China with a 45 km horizontal resolution and 28 vertical levels. The Goddard Global Ozone Chemistry Aerosol Radiation and Transport (GOCART) aerosol scheme (chem_opt =300) is adopted to simulate both dust and non-dust aerosol species (Chin et al., 2000, 2002). The Air Force Weather Agency (AFWA) dust emission scheme (dust_opt=3) is chosen for dust simulation. The mass mixing ratio of the main aerosol components, including dust, sea salt, organic carbon, black carbon, and sulfate, are predicted. Other main selected physics are identical to those of Dai et al. (2019). To better represent the characteristics of East Asian dust, the fractions of dust emissions in the AFWA scheme are modified to 0.034, 0.187, 0.327, 0.163, and 0.309 for the 0.2-2 μm (bin 1), 2-3.6 μm (bin 2), 3.6-6 μm (bin 3), 6-12 μm (bin 4), and 12-20 μm (bin 5) dust size bins in diameter, respectively (Su and Fung, 2015). To reduce the underestimation of dust emission in the AFWA scheme and start from a relatively unbiased simulation, the adjustable dust emission factor is

calibrated and selected as 21 based on the AERONET-observed AOT and AE. This rescaling of dust emission can benefit

the data assimilation since the Kalman filter assumes that the model is unbiased (Tsikerdekis et al., 2021). The aerosol optical properties are calculated with the Mie parameterization using the "Aerosol chemical to Aerosol Optical Properties" module (Ghan and Zaveri, 2007, Barnard et al., 2010), which is based on the sectional approach. The 8 dust size bins in Mie subroutine are the in the Model for Simulating Aerosol Interactions and Chemistry (MOSAIC) module. The calculation of the dust optical properties is improved with three corrections: (1) remap the fractions of AFWA bin 1 dust in 0.2-2 μm into

Mie calculation bins as Ukhov et al. (2021); (2) redistribute fractions of the dust mass based on the assumption that bin concentration is a function of natural logarithm radius as Ukhov et al. (2021); (3) increase the 8 dust size bins in Mie subroutine to 9 as 0.039-0.078, 0.078-0.156, 0.156-0.312, 0.312-0.625, 0.625-1.25, 1.25-2.5, 2.5-5.0, 5.0-10.0 μm, and 10-20 μm to distribute the AFWA bin 5 dust in 12-20 μm into bins for Mie calculation. To compare with AERONET observed aerosol optical properties, the simulated ones are calculated assuming that the particles are spherical and internally mixed

with all the simulated aerosol components (Barnard et al., 2010). The initial and lateral boundary meteorological conditions are from the NCEP Final (FNL) analysis. To reduce the uncertainties associated with the meteorological fields, the predicted wind (u, v), temperature (t), and specific humidity (q) by the WRF dynamical core are also nudged to the NCEP FNL analysis every 6 h for all layers (Dai et al., 2018).

## 3.2 Data assimilation framework

The adopted assimilation system, integrating measurements with model simulations, is based on the Ensemble Kalman Smoother (EnKS) with WRF-Chem (Dai et al., 2019). The Kalman Smoother is in essence a Kalman Filter that iteratively estimates emissions (Schutgens et al., 2012). As shown in Fig. S1, based on the EnKS, the dust emission of the WRF-Chem ensemble is optimized every 12 h, which corresponds to the assimilation time window of 12 h. Each assimilation cycle advances a time step of 12 h, and the dust emissions for 6 time steps are optimized using the observations in the 6[th] time step.

After each assimilation cycle, the dust emission for the first time step is the final optimized result, which has been optimized 6 times and will no longer be optimized in the next cycle. The finally optimized dust emissions therefore serve as the forcing dust emissions for advancing the system one time step, and they provide the initial conditions for the next assimilation cycle. The posterior dust emission $\bar{x}^a$ is obtained from the solution to the Kalman equations, as the following formula:

$$\bar{x}^a = \bar{x}^b + X^b \bar{w}^a, \tag{1}$$

where $\bar{x}^b$ and $X^b$ represent the ensemble mean of prior dust emission and the first guess ensemble perturbation, respectively. The weight matrix $\bar{w}^a$ determines the increment between the analysis and first guess as

$$\bar{w}^a = \tilde{P}^a (Y^b)^T R^{-1} (y^o - \bar{y}^b), \tag{2}$$

where the matrix $\tilde{P}^a (Y^b)^T R^{-1}$ is called the "Kalman gain"; $R$ is the observation error covariance matrix; the $y^o$ and $\bar{y}^b$ represent the assimilated hourly AOT and AE observations from AERONET and the first guess of the simulated AOT and

AE observations averaged over the ensemble members; the WRF-Chem serves as the observation operator $H$ to relate the

prior dust emission to the first guess of the simulated observations, $\bar{y}^b = H(\bar{x}^b)$; the first guess ensemble perturbation matrix in observation space $Y^b$ is calculated as $y^{b(i)} - \bar{y}^b$, $\{i = 1, 2, \ldots, k\}$ with k ensemble members. The analysis error covariance is obtained as

$$\tilde{P}^a = \left[(k-1)I + Y^{b^T}R^{-1}Y^b\right]^{-1}, \tag{3}$$

where I is the identity matrix. The analysis ensemble perturbations $X^a$ are obtained as

$$X^a = X^b W^a, \tag{4}$$

whose ith column is $x^a(i) - \bar{x}^a$, $\{i = 1, 2, \ldots, k\}$. In this study, the analysis ensemble by adding $\bar{x}^a$ to each of the columns of $X^a$ forms the optimal dust emission for the ensemble forecast to produce the initial conditions for the next analysis. $W^a$ is calculated as

$$W^a = [(k-1)\tilde{P}^a]^{1/2}. \tag{5}$$


This assimilation scheme offers the advantage of selectively assimilating observations for a given grid point by employing localization in the horizontal, vertical, and temporal scales (Hunt et al., 2007; Gaspari and Cohn, 1999; Miyoshi et al., 2007). The horizontal localization factor is calculated as $f(r) = exp(-r^2/2\sigma^2)$, the factor is truncated at 3.65 times the localization length $\sigma = 600$ km in this study, and $r$ is the distance between the observation and the grid centroid. The

vertical and time localization are not applied in this study.

### 3.3 Experimental design

To investigate the influences of AERONET AOT and AE assimilation on the dust emission optimization, three assimilation experiments are conducted from 12:00 UTC on 11 March 2021 to 00:00 UTC on 24 March 2021. The initial condition at 12:00 UTC on 11 March 2021 is prepared by an 11-day simulation executed by WRF-Chem without any aerosol data

assimilation as a spin-up. Due to WRF-Chem model uncertainties not only in dust emission but also in dust deposition (Huang et al., 2020) and dust optical properties (Di Biagio et al., 2019), two assimilation experiments with perturbation of dust emission and size distribution are conducted. One assimilation experiment named AOT DA-SZD assimilates only AERONET AOT observations, and the other assimilation experiment named AOT+AE DA-SZD assimilates both AERONET AOT and AE observations. 20 ensemble members are generated by perturbing the emission fluxes in each of the

five dust bins. The same perturbation factor is used across the whole domain. The random perturbation factor is drawn from a lognormal distribution with a mean of 1 and a standard deviation of 0.6. Refer to Dai et al. (2019), the standard deviation of 0.6 corresponds to the uncertainty of the dust emissions for 14 global models (Huneeus et al., 2011). Correlated noise is used across the dust size bins in the perturbation, and the noise correlation decreases with increased difference of the diameter among the bins (Di Tomaso et al., 2017). The ensemble prediction dynamically estimates the covariance between

the dust emission in each bin and the aerosol optical properties. The comparison between AOT DA-SZD and AOT+AE DA-SZD experiments shows the effects of the additional AE information on dust emission optimization. The effects of dust emission size distribution perturbation are investigated by one additional assimilation experiment named AOT+AE DA,

which is conducted as the AOT+AE DA-SZD experiment except that the 20 ensemble members are generated by perturbing the dust emission in each bin with the same perturbation factor. The results from 12:00 UTC on 11 March 2021 to 23:59 UTC on 13 March 2021 are excluded in the analysis. The baseline experiment named FR is a single run. It does not assimilate any observations but otherwise share the same configuration with the assimilation experiments.

## 4 Results

### 4.1 Dust emission and simulated dust field

Figure 2 shows the temporal variations of hourly accumulated dust emissions in the Taklimakan desert and Gobi desert simulated by FR, AOT DA-SZD, AOT+AE DA-SZD, and AOT+AE DA experiments. The first dust storm in the Taklimakan desert mainly emits from 00:00 UTC on 14 March to 12:00 UTC on 16 March, and reaches the peak value of 420, 887, 952, and 933 kt h$^{-1}$ at 01:00 UTC on 16 March for the FR, AOT DA-SZD, AOT+AE DA-SZD, and AOT+AE DA experiments, respectively. The second dust storm in the Taklimakan desert mainly emits from 00:00 UTC on 17 March to 20:00 UTC on 18 March, and reaches the peak value of 1033, 1914, 1272, and 1338 kt h$^{-1}$ at 07:00 UTC on 18 March for the FR, AOT DA-SZD, AOT+AE DA-SZD, and AOT+AE DA experiments. There are almost no dust emissions in Taklimakan desert after 20:00 UTC on 18 March. Regarding the Gobi desert, the strongest dust storm generally emits from 00:00 UTC on 14 March to 18:00 UTC on 15 March, and reaches the peak value of 1735, 3253, 3837, and 2791 kt h$^{-1}$ at 07:00 UTC on 14 March for the FR, AOT DA-SZD, AOT+AE DA-SZD, and AOT+AE DA experiments. After 18 March, there are five relatively weak dust processes in the Gobi desert. Depend on the temporal variations of dust emission in the Gobi desert, the whole period is divided into two dust processes: the strong dust storm from 14 March to 17 March 2021 and the weak dust storm from 18 March to 23 March 2021. In general, the total dust emissions during 14-17 (18-23) March 2021 in Gobi desert are 21.45 (13.03), 32.03 (18.84), 44.41 (8.22), and 33.10 Tg (8.57 Tg) for the FR, AOT DA-SZD, AOT+AE DA-SZD, and AOT+AE DA experiments. The dust emission in each dust bin is given in Table S1.

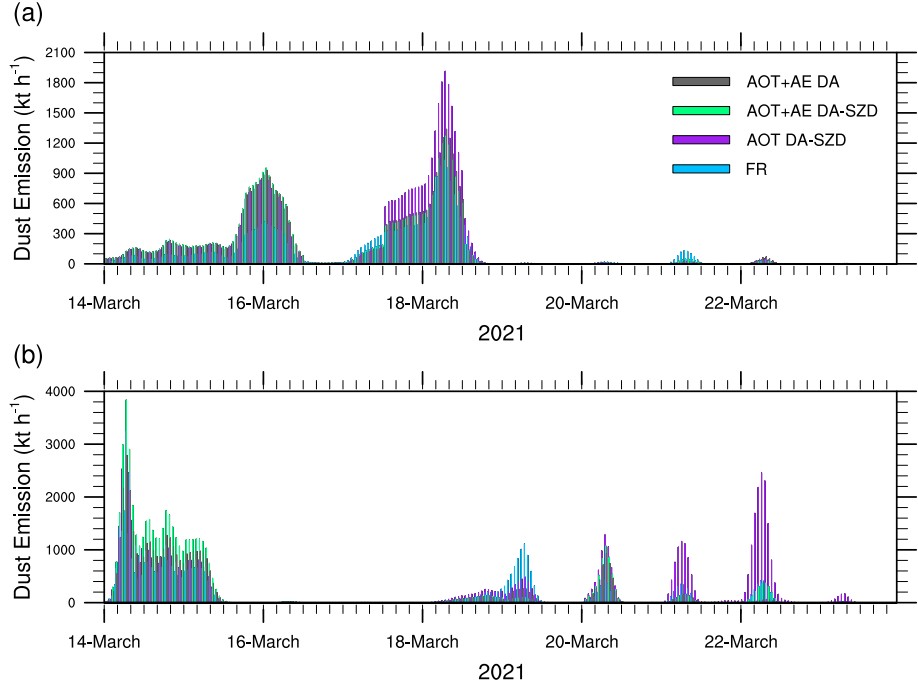

**Figure 2. Time series of hourly accumulated dust emissions (units: kt h⁻¹) for FR, AOT DA-SZD, AOT+AE DA-SZD, and AOT+AE DA experiments during 14-23 March 2021 summed across the Taklimakan desert (a) and Gobi desert (b) (marked in Fig. 3).**

The simulated accumulated dust emission, averaged AOT and AE for the four experiments during 14-17 March 2021 are given in Fig. 3. During 14-17 March 2021, the dust storm in the Taklimakan desert and Gobi desert both transports eastward, mainly affecting Northern China. The dust emissions in the three assimilation experiments are significantly higher than that in FR experiment, especially AOT+AE DA-SZD experiment. The differences of dust emission between the FR and assimilation experiments are mainly concentrated in the Taklimakan desert and Gobi desert, while there is generally no difference in India due to the distance truncation of horizontal localization. The increased dust emissions in all the assimilation experiments induce higher AOTs over the deserts and the associated downwind regions, whereas only the AEs over the Gobi desert and its downwind region in the AOT+AE DA-SZD experiment are significantly reduced. The ratios between posterior error of dust emission and the prior one in each dust bin for the three assimilation experiments during 14-17 March 2021 are shown in Fig. 4. This helps visualize how adding AOT and AE for the data assimilation reduces the posterior errors of simulated dust emissions. Due to the same prior error of the dust emission, the difference in ratio represents the difference in posterior error. A ratio with a value lower than 100% indicates the assimilation decreasing the uncertainties of the dust emission, and a lower value represents higher constraint. The posterior error increases with the dust size bin increased in the AOT DA-SZD experiment, which is due to the time-lagged AOT observations in the downwind areas are more relevant with the fine-mode dust emission because of the stronger gravity settling of coarse-mode dust and

the higher extinction efficiency of fine-mode dust (Fig. S2 and Fig. S3). The additional AE observations in the AOT+AE DA-SZD experiment further adjust the dust emission size distribution over the Gobi desert by decreasing dust emission in

bin 1 and increasing dust emission in bin 3 (Fig. S4 and Fig. S5), inducing significant reductions of the posterior error in bin 1 and bin 3 in Gobi desert. The significant decrease in bin 1 is due to bin 1 having the highest dust extinction efficiency, and the significant increase in bin 3 is due to bin 3 having the largest proportion in the fine-mode dust emission. This induces the significant decrease of the AE over the Gobi desert and its downwind region due to the assimilation of additional AE observations can obviously modify the normalized atmospheric dust volume distribution (Fig. S6). Although AOT+AE DA

experiment includes the AE observations, there is no changes in the dust emission size distribution due to the same perturbation parameter (Fig. S4 and Fig. S5). This leads to the simulated AEs are similar as the FR experiment. It is found that dust emission in bin 3 over the Gobi desert in the AOT+AE DA-SZD experiment is obviously higher than that in the other assimilation experiments (Table S1), however, there is no significant difference in AOT among the three assimilation experiments. This is due to the effect of increased emission in bin 3 on AOT being offset by the effect of decreased emission

in bin 1, since the coarse-mode dust is generally removed by gravitational sedimentation (Fig. S2) and the coarse-mode dust has lower extinction efficiency (Fig. S3).

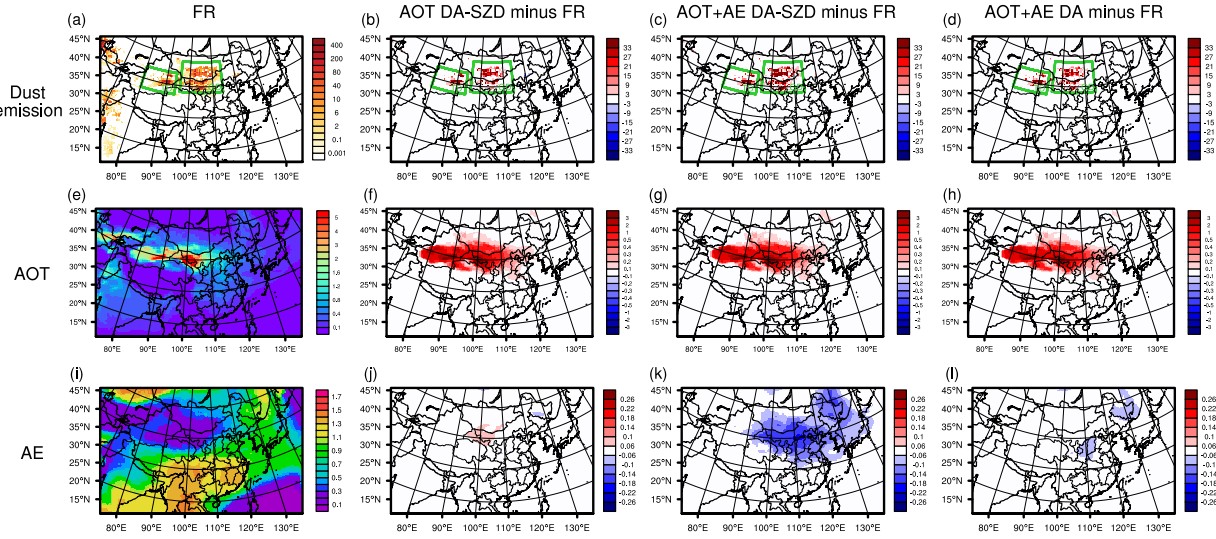

**Figure 3. Spatial distributions of accumulated dust emission, aerosol optical thickness (AOT), and Angstrom Exponent (AE) for FR experiment during 14-17 March 2021 (a,e,i). Differences of accumulated dust emission (b,c,d), AOT (f,g,h), and AE (j,k,l)**

**between AOT DA-SZD, AOT+AE DA-SZD, and AOT+AE DA experiments minus FR experiment. The unit of dust emission is g m$^{-2}$. The green boxes represent the Gobi desert (GD) and Taklimakan desert (TD).**

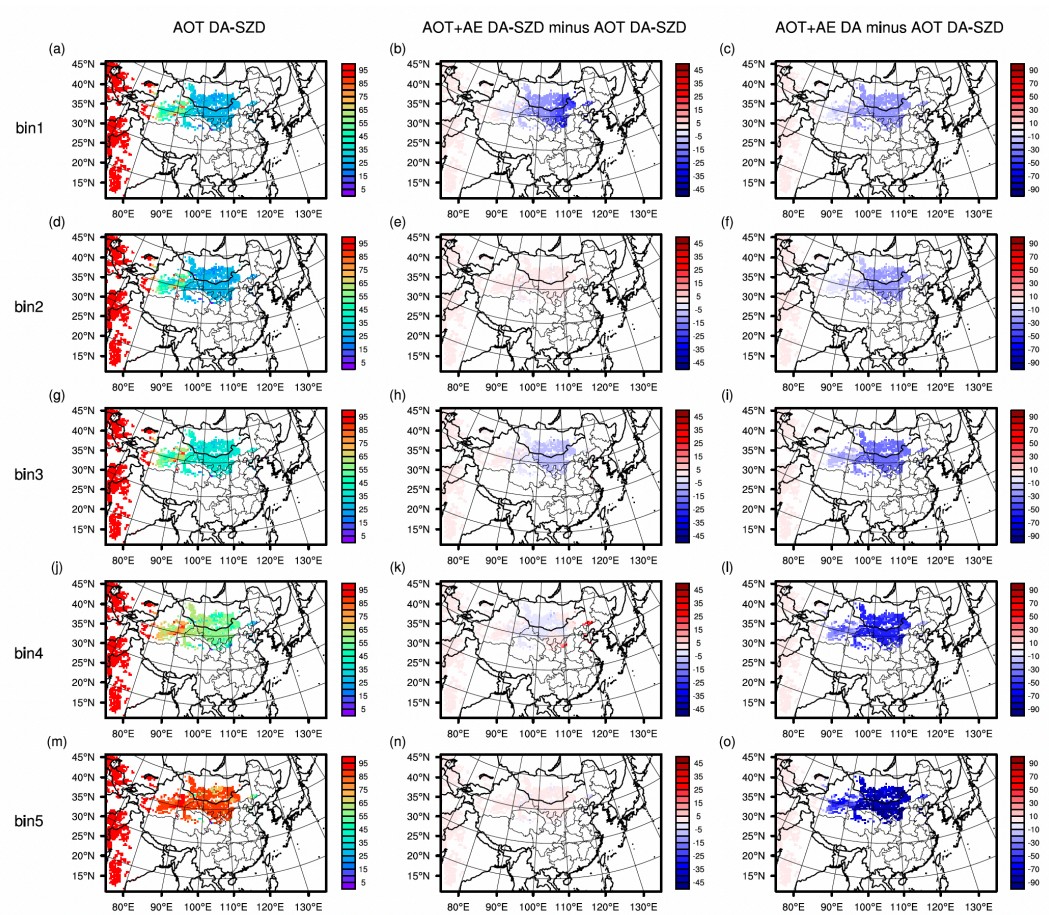

**Figure 4. The ratio between posterior error of FR simulated dust emission and prior error of optimized dust emission in the 5 dust size bins for AOT DA-SZD experiment (a,d,g,j,m) during 14-17 March 2021. Differences of the ratios between AOT+AE DA-SZD (b,e,h,k,n) and AOT+AE DA (c,f,i,l,o) experiments minus AOT DA-SZD experiment.**

The simulated accumulated dust emission, averaged AOT, and AE for the four experiments during March 18-23 2021 are given in Fig. 5. Compared with the FR experiment, the dust emission in the AOT DA-SZD experiment is increased in Gobi desert of Mongolia and decreased in Gobi desert of China while the dust emission in the AOT+AE DA-SZD and AOT+AE DA experiments are decreased in most part of Gobi desert. Compared with the FR experiment, the dust emission in the AOT+AE DA-SZD experiment is increased over the Taklimakan desert and some regions of the Gobi desert but decreased over the majority of the Gobi desert. This is because the dust emission periods vary across different grids, leading to the opposite trends among different dust source regions. Due to the southward transport of part of the dust emitted from the Gobi desert, the decreased dust emission induces lower AOTs in Southern China. It is also found that only the AEs over the downwind region of the Gobi desert in the AOT+AE DA-SZD experiment are significantly reduced. As given in Table S1, the AOT DA-SZD experiment significantly increases the dust emission in bin 5 and slightly decreases the dust emissions in

bin 1, bin 2, bin 3, and bin 4 over the Gobi desert, whereas the AOT+AE DA-SZD and AOT+AE DA experiments with the additional AE observations decrease the dust emission in each bin over the Gobi desert. The ratios between the posterior error of dust emission and the prior one in each dust size bin for the three assimilation experiments during 18-23 March 2021 are shown in Fig. 6. It is interesting that the ratios in the AOT DA-SZD and AOT+AE DA-SZD experiments during 18-23 March 2021 are lower than those during 14-17 March 2021. This is due to the lower AOT observations and associated observation errors at the dust source site Dalanzadgad during 18-23 March 2021 generate more constraints on dust emission than those during 14-17 March 2021. AOT DA-SZD experiment significantly increases the dust emission in bin 5 and leads to its emission fraction exceeding 70% (Fig. S4), however, this phenomenon is not found in AOT+AE DA-SZD experiment. The obviously lower posterior error in bin 5 in the AOT+AE DA-SZD experiment indicates that the additional AE observations can eliminate the sharp increase of dust emission in bin 5 and constrain the dust emission size distribution with higher confidence.

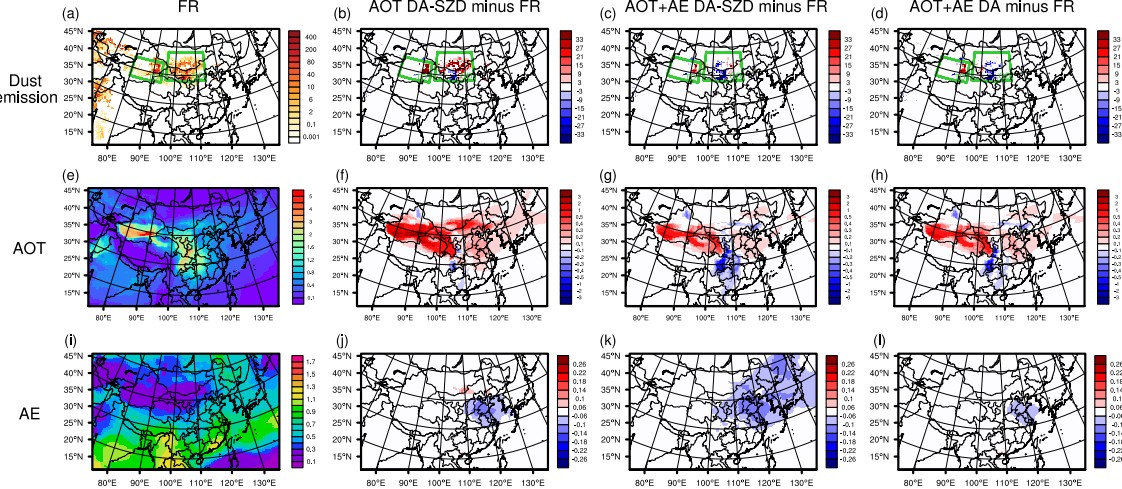

**Figure 5. Same as Fig. 3 but during 18-23 March 2021. The unit of dust emission is g m$^{-2}$.**

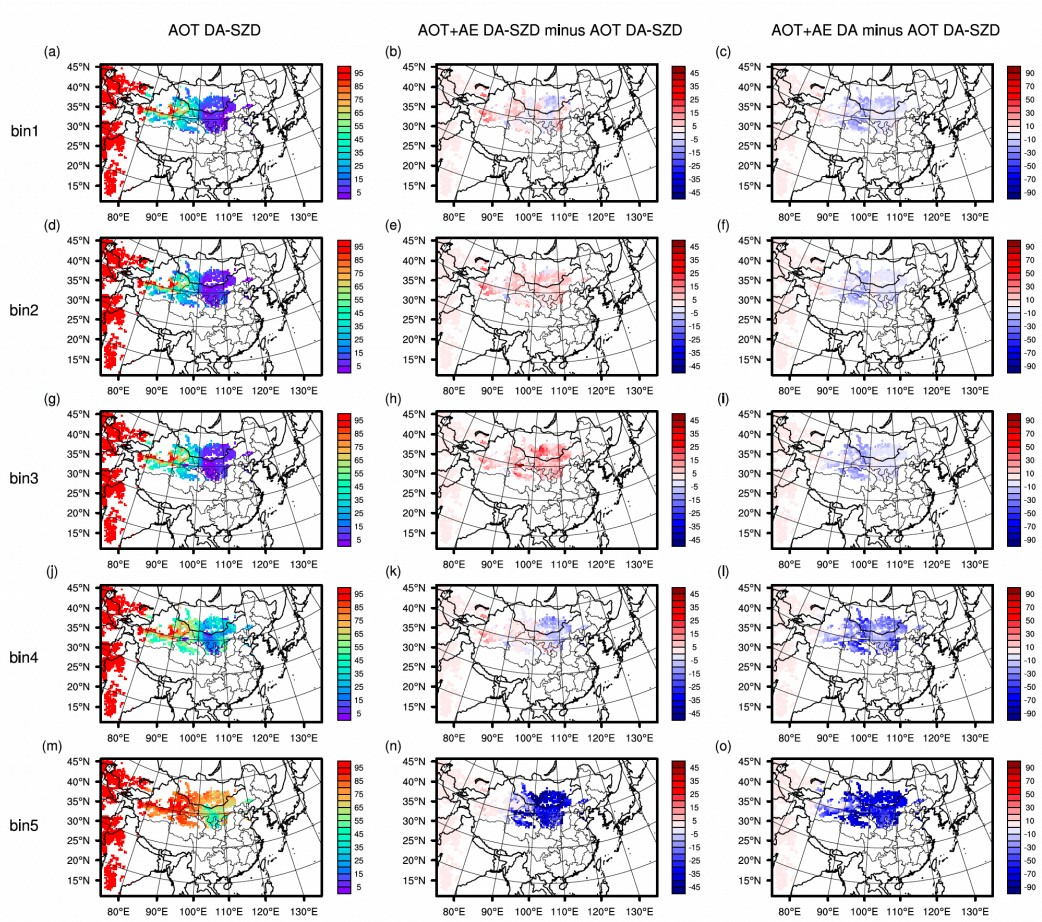

 **Figure 6. Same as Fig. 4 but during 18-23 March 2021.**

## 4.2 Evaluation of simulated AOTs and AEs

To validate the posterior dust emission, the simulated hourly AOTs and AEs with the prior and posterior dust emissions are compared with the assimilated AERONET observations as the sanity check in Fig. 7. The comparisons with independent SONET observations are further given in Fig. 8. To quantify the model performances, statistical criteria (Boylan and Russell, 2006; Willmott et al., 2012; Yumimoto et al., 2017), including the mean bias (BIAS), the mean fractional error (MFE), the root mean square error (RMSE), and the index of agreement (IOA) are calculated between the simulated results and observations. It is apparent that all the three assimilation experiments can optimize the dust emissions to better simulate AOTs and AEs closer to the assimilated AERONET and independent SONET observations. Compared to the independent SONET observations, the AOT DA-SZD experiment reduces the AOT BIAS and RMSE of the FR experiment by 85% and 14%, however, it can only reduce the AE BIAS and RMSE of the FR experiment by 4% and 1%. This indicates the assimilation of AERONET AOT observations can only optimize the dust emission but not the dust emission size distribution.

The AOT+AE DA-SZD experiment reduces the AOT BIAS and RMSE of the FR experiment by 92% and 17%, and it can also reduce the AE BIAS and RMSE of the FR experiment by 68% and 62%. Although the AOT+AE DA experiment assimilates the AE observations, however, it has limited improvement of AE since the uncertainty of dust emission size distribution is not considered. These results indicate the additional assimilation of AE observations with consideration of the dust emission size distribution uncertainty are helpful for the optimization of dust emission through the adjustment of dust size distribution. Similar conclusions are also found in the comparison with the assimilated AERONET observations.

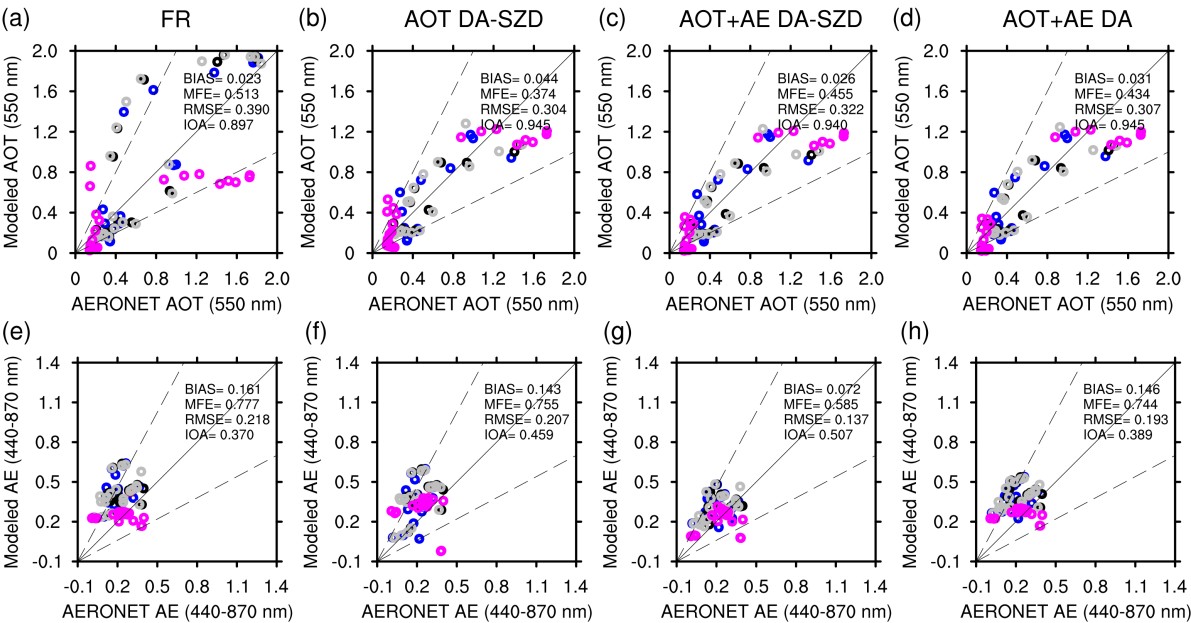

**Figure 7. Scatter plots of assimilated AErosol RObotic NETwork (AERONET) hourly aerosol optical thicknesses (AOTs) versus the simulated ones at 550 nm for FR (a), AOT DA-SZD (b), AOT+AE DA-SZD (c), and AOT+AE DA (d) experiments from 14 March to 23 March 2021. The circles with different colours represent different AERONET sites: Beijing-CAMS (dark blue), Beijing (black), Beijing_PKU (brown), Beijing_RADI (grey), and Dalanzadgad (violet). The solid black line is the 1:1 line and the dashed black lines correspond to the 1:2 and 2:1 lines. BIAS, MFE, RMSE, and IOA represent the mean bias, the mean fractional error, the root mean square error, and the index of agreement. (e,f,g,h) Same as (a,b,c,d) but for Angstrom Exponents (AEs) in the wavelength 440-870 nm.**

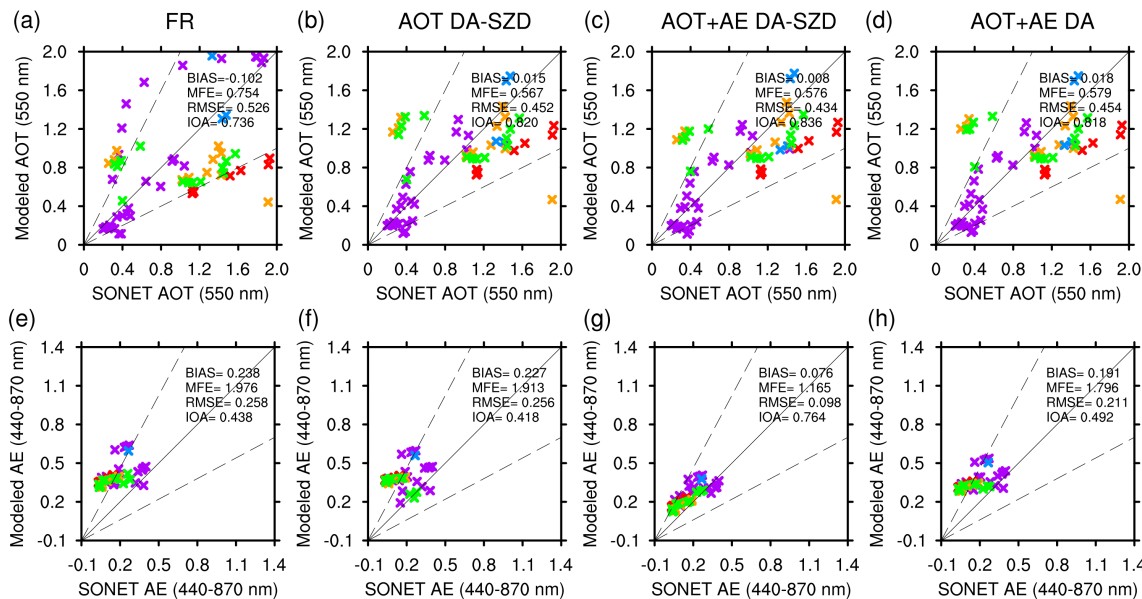

**Figure 8. Same as Fig. 7 but for Skynet Observation NETwork (SONET) hourly observations for independent validation. The crosses with different colours represent different SONET sites: Yanqihu (light blue), Beijing (purple), Jiaozuo (red), Songshan (orange), and Zhengzhou (green).**

Time series of the simulated AOT and AE at AERONET and SONET sites for the four experiments are further given in Fig. 9 and Fig. 10, respectively. The BIAS and RMSE of simulated and observed AOT and AE over each site are given in Table S2 and Table S3. It is found that the FR experiment can generally reproduce the temporal variations of the observed aerosol optical properties especially AOT over all sites, indicating the dust processes simulated by WRF-Chem are reasonable. The reasonable simulated dust processes demonstrate the covariance between the simulated dust emissions and aerosol optical properties is reliable for optimizing the dust emission. Due to the FR experiment underestimating AOTs and overestimating AEs over both the dust source and downwind sites during 16-17 March 2021, the assimilation of only AOT observation leads to an increase in the dust emission in each bin except bin 4 over the Gobi desert during 14-15 March 2021 (Table S1) and the additional AE observation leads to a significantly decrease of dust emission in bin 1 and an increase of dust emission in bin 3. The latter induces the simulated AEs on March 16 to be comparable to the observed ones especially over the independent SONET sites named Songshan and Zhengzhou, and this proves that not only the simulated dust emissions but also their size distributions over the Gobi desert in the AOT+AE DA-SZD experiment are optimized. The superiority of the adjustment of dust emission size distribution is further demonstrated by the limited effects of AE observation on the model in the AOT+AE DA experiment due to the uncertainty of dust size distribution not being considered. Due to the simulated AOTs during 20-23 March 2021 in the FR experiment being underestimated except on March 21 in dust source site and overestimated in downwind sites, the assimilation of AOT leads to slightly decreasing the dust emissions in bin 1, bin 2, bin 3, and bin 4 and significantly increasing in bin 5 (Table S1). This induces the increase of the total dust emission in the Gobi

desert of Mongolia and decrease of the total dust emission in Gobi desert of China (Fig. 5). Due to the simulated AEs during 20-23 March 2021 in FR experiment being slightly underestimated except on March 23 in the dust source site and

350 significantly overestimated in the downwind sites, the assimilation of additional AE induces the decrease of dust emission in each bin.

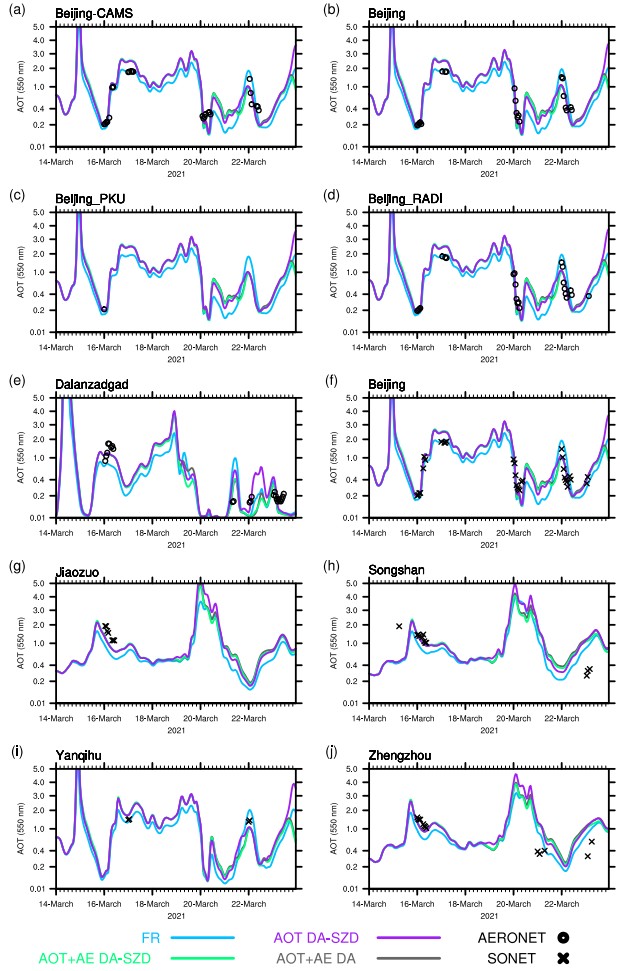

**Figure 9. Hourly time series of the simulated aerosol optical thicknesses (AOTs) for the FR, AOT DA-SZD, AOT+AE DA-SZD, and AOT+AE DA experiments and the observed ones over AERONET sites (a-e) and SONET sites (f-j) during 14-23 March 2021.**

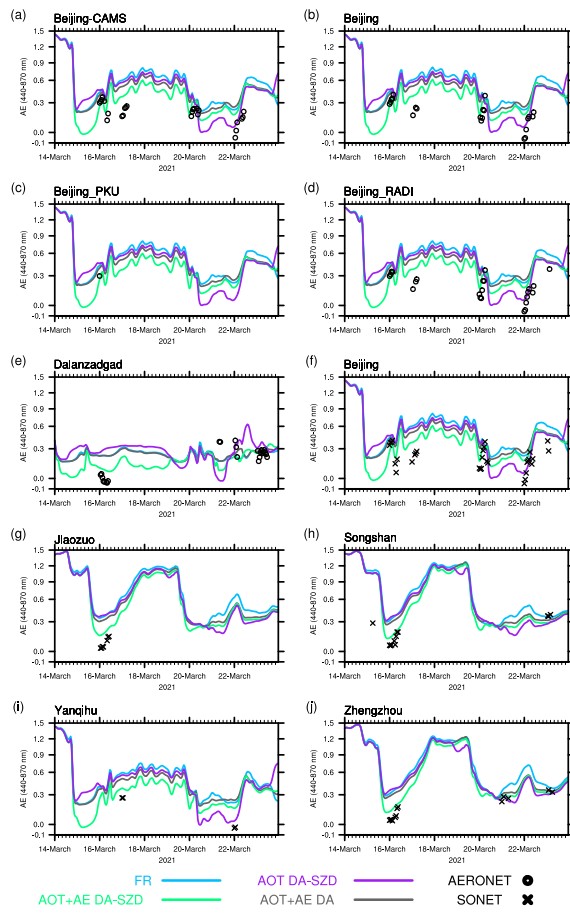

355

**Figure 10. Same as Fig. 9 but for Angstrom Exponents (AEs) in the wavelength 440-870 nm.**

### 4.3 Evaluation of simulated aerosol vertical extinctions

To further evaluate the dust emission optimization, the simulated aerosol extinction coefficients are compared with the independent CALIOP observed ones in the three CALIPSO orbit paths (Fig. 11). Due to the AOT+AE DA-SZD experiment has the best performance among the three assimilation experiments, the results for the FR and AOT+AE DA-SZD experiments in the three CALIPSO orbit paths are given in Fig. 12. Two paths near the dust source region cross the westward pathway of dust transport at 19:18:09 UTC on 15 March (path A) and the eastward pathway of dust transport at 05:55:25 (UTC) on 16 March (path B), and one path is far away from the dust source region at 18:17:32 UTC on 16 March (path C) (Fig. S9). As shown in Fig. 12, the vertical aerosols in path A are dominated by dust from 1 km to 12 km altitude. The FR experiment can capture the observed aerosol vertical patterns, whereas it significantly underestimates the aerosol extinction coefficients near the surface. The AOT+AE DA-SZD experiment reduces the underestimation and performs a more reasonable magnitude of aerosol extinction coefficients with values higher than 1 km$^{-1}$ around the surface from 35°N to 41°N and 0.1 km$^{-1}$ from 2 km to 4 km around 40°N. This indicates that the assimilation of AOT and AE observations can

better reproduce the features of aerosol vertical variations during dust transportation near the dust source region. In addition, it should be noted that the improvements of the aerosol extinctions with posterior dust emission on March 15 benefit from assimilating the time-lagged observations from downwind areas. In path B, the dust transported from west to east is mainly concentrated at 4 km and the FR experiment generally reproduces this dust vertical structure with significant underestimations. The AOT+AE DA-SZD experiment can improve the underestimations and the simulated aerosol extinctions are further consistent with the observed ones. CALIOP observed aerosol extinctions from 6 to 7 km around 41°N are higher than 1 km$^{-1}$, while the simulated aerosol extinctions for the FR experiment are around 0.3 km$^{-1}$. The AOT+AE DA-SZD experiment successfully reproduces the magnitude and variations of aerosol extinctions around 41°N. The vertical aerosols in path C are also dominated by dust in all heights. Although the FR experiment successfully reproduces the vertical structure of double dust-layers observed by CALIOP between 33°N to 43°N, the aerosol extinction coefficients are significantly underestimated around 4 km. The AOT+AE DA-SZD experiment increases the transported dust, diminishing the underestimation of aerosol extinction coefficients. The aerosol extinction coefficients in the AOT+AE DA-SZD experiment are more comparable to the CALIOP observations. This proves that the assimilation of AOT and AE observations can also better reproduce the features of dust vertical distributions in areas far away from the dust source region.

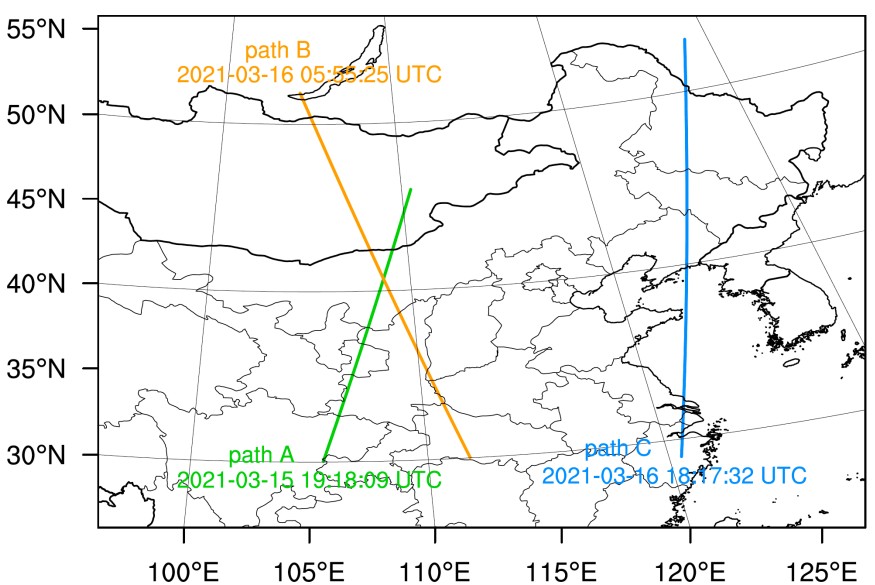

**Figure 11. The Cloud-Aerosol Lidar and Infrared Pathfinder Satellite Observations (CALIPSO) 3 orbit paths during 14-23 March 2021.**

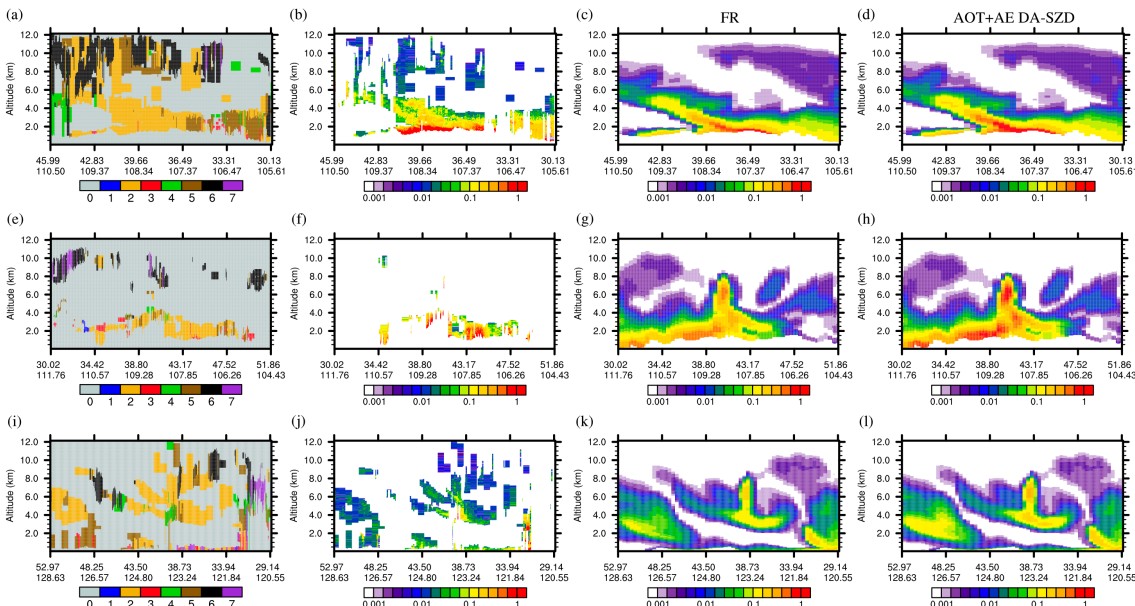

**Figure 12. (a) Time-height cross section of CALIPSO-derived vertical aerosol subtypes in path A. The Cloud-Aerosol Lidar with Orthogonal Polarization (CALIOP)-observed aerosol extinction coefficients at 532 nm (km$^{-1}$) (b) and the simulated ones at 550 nm in the FR (c) and AOT+AE DA-SZD (d) experiments. (e-h) Same as (a-d) but for path B. (i-l) Same as (a-d) but for path C.**

## 5 Conclusions

To investigate the additional benefit of aerosol size information in dust emission optimization, the Aerosol Robotic Network (AERONET) ground-based aerosol optical thickness (AOT) and Ångström Exponent (AE) time-lagged observations are assimilated during the severe East Asian dust storm outbreak in March 2021 in this study. The Ensemble Kalman smoother (EnKS) assimilation framework (Dai et al., 2019) with the Weather Research and Forecasting model coupled with Chemistry (WRF-Chem) version 4.4 is applied for the dust emission optimization.

Three assimilation experiments are conducted during 14-23 March 21. The first one named "AOT DA-SZD" only assimilates AERONET AOT observations with perturbation of dust emission and size distribution, the second one named "AOT+AE DA-SZD" assimilates is conducted as same as the first one except assimilating both AERONET AOT and AE observations, and the third one named "AOT+AE DA" is conducted as same as the second one except the ensemble members are generated by perturbing the dust emission in each bin with same perturbation factor. The baseline experiment "FR" without assimilation is used for comparison.

Our results demonstrate that the additional assimilation of AE observations with consideration of the dust emission size distribution uncertainty are helpful to the optimization of dust emission through better adjustment of dust size distribution. AOT assimilation can only optimize the dust emission flux depending on the covariance between time-lagged AOT observations and the simulated total dust emission, while the additional inclusion of AE assimilation can optimize the size

distribution of dust emission and the associated total flux depending on the covariance between time-lagged AE observations and the simulated dust emission in each bin.

All three assimilation experiments can optimize the dust emissions to better reproduce the assimilated AERONET and independent SONET AOT and AE observations. Although the assimilation of AOT observations can only optimize the dust emission but not the dust emission size distribution, the assimilation with the additional of AE observations can not only reduce the AOT BIAS and RMSE of the FR experiment by 92% and 17%, but also reduce the AE BIAS and RMSE of the FR experiment by 68% and 62% through optimizing both the dust emission size distribution and the associated total flux. The temporal variation of simulated AOT and AE can also be improved through assimilating additional AE information. The assimilation of AOT and AE also makes the magnitude and variations of aerosol vertical extinctions more comparable to independent Cloud-Aerosol Lidar with Orthogonal Polarization (CALIOP) observations in both the westward and eastward pathways of dust transport.

This study emphasizes the additional AE assimilation is useful in the dust emission optimization. To further explore the roles of the assimilated observations on the dust emission optimization and accurate simulation of dust life cycle, sensitivity experiments should be conducted to quantify the influences of observation uncertainties and frequencies on the assimilation efficiency. The assimilation parameters such as spatial and temporal localization length are also important for dust emission optimization. In addition, the coarse-mode AERONET AOT from the spectral deconvolution algorithm (SDA) is also useful for dust emission optimization since all fine-mode aerosols are truncated and only dust/sea-salt remains.

**Code and data availability**

All data used in this study is freely available from public data repositories. AERONET products are available from https://aeronet.gsfc.nasa.gov/new_web/download_all_v3_aod.html. SONET products are available from http://www.sonet.ac.cn/en/cpin/html/?194.html. CALIOP products are available from the NASA Langley Research Center–Atmospheric Sciences Data Center (ASDC).

**Author contributions**

YC conceived the study and designed the dust storm data assimilation. YC and TD performed the control and assimilation tests and carried out the data analysis. JC, DG, JJ, TN, and GS provided useful comments on the paper. YC prepared the manuscript with contributions from TD and all others co-authors.

**Acknowledgments**

This study was financially supported by the National Natural Science Funds of China (grant nos. 42175186, 42305088, 42375190), the China Postdoctoral Science Foundation (grant no. 2022M723091), the Special Research Assistant Project of the Chinese Academy of Sciences, the Open fund by Jiangsu Key Laboratory of Atmospheric Environment Monitoring and Pollution Control (KHK 2206), the Key Laboratory of Atmospheric Chemistry, China Meteorological Administration,

LAC/CMA (grant no. 2022B05), the Youth Innovation Promotion Association CAS (grant no. 2020078), and the International Partnership Program of Chinese Academy of Sciences (grant no. 134111KYSB20200006). Model simulations were performed using NEC SX-Aurora TSUBASA supercomputers at NIES, Japan. We thank to the relevant researchers who provided AERONET, SONET, and CALIOP observations. We also thank the anonymous reviewers for their valuable comments and suggestions that improved the manuscript.

**Competing interests**

The contact author has declared that none of the authors has any competing interests.

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
