# Peer review of "Improving estimation of a record breaking East Asian dust storm emission with lagged aerosol Ångström Exponent observations"

_EGUsphere, 2024_

## Referee Comment (RC1)

The manuscript, titled "Improving estimation of a record breaking East Asian dust storm emission with lagged aerosol Ångström Exponent observations", by Yueming Cheng et al. is a data assimilation (DA) case study for improving dust storm predictions in China using WRF-Chem. The authors chose a dust storm in Mar 2021 as a case study and successfully showed that using AERONET observations for DA could improve hindcasting ability of dust in WRF-Chem. Authors further showed that employing the Angstrom Exponent (AE) benefits more than employing the aerosol optical thickness (AOT) in improving DA results. The methodology is generally scientifically valid and clear in presentation, although there are occasional grammar issues or unclear descriptions that could use a little more editing. The main issue I thought was that there are insufficient science discussions to the results, such as why AE benefits more than AOT, or how could the WRF-Chem dust model be improved. The manuscript so far feels a little more like a technical report on improving hindcasts rather than a scientific development on our understanding of dust modelling. I have a few major comments below for science, and I have some other specific comments on technical questions or presentation. I suggest major revision for the current revision.

Major comments:

- I think there needs to be some science discussions on what caused the biases in WRF-Chem dust emissions in China. The paper currently leaves readers with puzzles regarding why WRF-Chem underestimates dust so much. Is it problems in simulated dust emissions, lifetime (dust deposition), or optical properties? If the bias comes from emission, is it a problem in wind speed, soil moisture, vegetation, or other met fields? If it's deposition or optics, is there anything to do with size distribution, dust particle shape, or dust refractive index? Does the dust underestimation also occur over the rest of the world in WRF-Chem? How does changing the *a priori* emissions (using other dust_opt or other emission schemes) alter the FR underestimations? Modellers would like to know how could our process-based dust understanding benefit from the insights from this DA study.

- It looks like authors attributed all the differences/biases between AERONET-measured and WRF-simulated AOT to dust. Could the biases be attributed to other natural and anthropogenic emissions? Although you only assimilated three days where dust was dominant, there must be some strong anthropogenic and natural emissions that got captured by AERONET, especially over Beijing, a heavily polluted metropolitan area. You only used AE values < 0.6 for evaluation to focus on dust, but it seems you didn't do the same when doing the DA. From my point of view, it could be better to use the coarse-mode AERONET AOT from the spectral deconvolution algorithm (SDA) to do the DA since all fine-mode aerosols are truncated and only dust/seasalt remains.

- Even if all biases in AOT/AE were assumed to be due to dust, currently the authors attribute all differences/biases between observations and simulations to emissions and only correct emissions. This presumes there are no biases in dust settling/deposition and dust optical properties. But if so, assimilating AOTs should correct most of the error, and AE would not be needed. A science discussion is needed on why the AE information could further reduce bias. Authors could not just conclude that the more you use for DA the better the hindcast results. My thought is that if AE is additionally needed for DA, this either means

there are problems in AERONET AOT, or (more likely) WRF-Chem has problems not only on dust emissions but also on dust optics. Studies also have pointed out issues in both settling velocity (e.g., Huang et al., 2020) and optics (e.g., Di Biagio et al., 2019) in models. If so, it does not make so much sense to attribute all AOT/AE biases to emissions to compensate other errors in WRF-Chem. Maybe optics should also be inverted, not just emissions.

Other specific comments:

Line 78: It's a little difficult to grasp how many AERONET stations you used for assimilation. Please state in text. I suggest plotting out the locations of the AERONET stations, with values of AOT and AE, either in Fig. 1 or in SI.

Lines 87-88: Authors wrote two observation errors: "observation error is a sum of representation error and observation error", which is confusing. Maybe use another word like instrument error for the latter one?

Line 89: I glanced through Schmid 1999 but didn't see such characterization of representation error. Schmid 99 was not a WRF-Chem modelling study either. Why should representation error be 0.055τ? Should it be more related to WRF-Chem grid resolution? Also, please define τ.

Line 92: I think it is needed to state why authors chose to assimilate AERONET instead of SONET or CALIOP. It looks random to me.

Line 93: Please also plot out the locations of the SONET sites and their values of AOT and AE, in main text or supplement.

Line 117: Please define the acronym MOSAIC. I am not sure how important this modification is if you don't concern dust chemistry, since MOSAIC never appeared in the text again. How are the spatiotemporal distributions of metal ions changed through this modification?

Line 140: Again, it looks like the whole difference between AERONET AOTs and simulated AOTs are attributed toward dust emission biases. Can't there be biases from other natural and anthropogenic emissions?

Line 145: I am not sure if the error covariance includes forward model error (that is, error in the WRF-Chem H operator). Please clarify.

Line 155: A localization length of 600 km sounds a little too big to me for assimilating AERONET data. It is almost a meso-synoptic length scale and is much bigger than your WRF-Chem horizontal grid resolution.

Line 155: "Grid centroid" instead of "centre grid"?

Figure 1: It's interesting that using AE measurements in Beijing could lead to changes in the posterior AE (AOT+AE DA) over Taklimakan, or even in India, in comparison to the *a priori* AE

(FR). How does DA generate emission changes in Taklimakan and India, if you were using AERONET sites in Beijing?

Line 203-206: It is not clear what this means. There were no observations because AERONET sites were down, like because of the dust storm? Please rephrase. Does this mean if you use the observations on 14-15 March, dust emissions would be even higher? Please clarify.

Line 208: I suggest adding map plots on the prior error of the WRF FR emissions, as well as the posterior errors of inverted emissions for the AOT DA and the AOT+AE DA cases. It helps visualize how adding AOT and AE for DA reduces the posterior errors of DA emissions.

Figure 4: When you say "aggregate", is this plot averaged across or summed across the domain? Please clarify in text.

Line 213: I suggest either saying posterior and prior dust emissions, or *a posteriori* and *a priori* dust emissions.

Lines 217-219: So, did you use more AERONET sites than listed here for DA above? Or are these all the sites used for DA? I am still confused, please clarify.

Line 236: Here authors should suggest scientific reasons for why using AE would benefit DA so much, while AOT DA would not.

Line 248: What's the reason of selecting these two SONET sites but not the other two? It looks a little random here. I suggest plotting the comparisons for Songshan and Jiaozuo in main text or supplement too.

Line 255: From here on, I start to find the message for the next few subsections a bit repetitive, stating that the AOT+AE DA run is better than the FR run and the AOT DA run. The manuscript could use a little rewriting to make the discussion and message more succinct.

Figure 6: It seems to readers there are insufficient SONET data points for the time series plot. I suggest authors also include SONET data points for AE > 0.6 since you used it for DA, in any color other than grey.

Figure 7-9: These are nice plots. Though, readers find the message across Figs. 7-9 a little repetitive. I would suggest showing the extinction coefficient (second rows) and skip the AOT plots (first rows), and maybe combine second rows of all three figures together. The first rows could be put in supplement.

---

## Author Comment (AC1)

**Responses to the Community comment**

We would like to thank to Dr. Alexander Ukhov, for giving constructive criticisms, which are very helpful in improving the quality of the manuscript. We have made revision based on your critical comments and suggestions. The community comments are reproduced (*black*) along with our replies (blue) and changes made to the text (red) in the revised manuscript. All the authors have read the revised manuscript and agreed with submission in its revised form.

**Comment NO.1:** *I would advice authors to switch to simple dust_opt=1 as it is known that the AFWA scheme strongly underestimates dust emissions.*

**Response:** Thank you very much for your advice. As recommended by RC2, to reduce the underestimation of dust emission in AFWA scheme and start from a relatively unbiased simulation, the adjustable dust emission factor is calibrated and selected as 21 based on the AERONET-observed AOT and AE.

In order to investigate the influence of different dust emission schemes on the dust emission during 14-23 March 2021, another experiment named FR-dustopt1 (dust_opt=1) has been conducted to compare with FR experiment (dust_opt=3). The accumulated dust emissions in Gobi desert during 14-23 March 2021 are 36.99Tg and 43.00Tg for FR and FR-dustopt1 experiments. In general, the dust emission simulated by GOCART and AFWA dust emission schemes have no significant differences in Gobi desert. Therefore, we still use AFWA dust emission scheme in the revised manuscript.

**Changes in Manuscript:** Please refer the description in the revised manuscript, from Page 5 Line 144 to Page 6 Line 146.

**Comment NO.2:** *Also I could not find which WRF-Chem version has been used. I recommend v 4.1.3 and above.*

**Response:** Done. We have modified the WRF-Chem version from 3.5 in the original manuscript to 4.4 in the revised manuscript.

**Changes in Manuscript:** Please refer the description in the revised manuscript, Page 5 Line 136.

**Comment NO.3:** *Also there is a bug in the calculation of optical properties when GOCART scheme is used, i.e. 5th dust bin is not accounted. Authors are welcome to contact me if it is needed.*

**Response:** Done. We have already fixed the bug in the calculation of optical properties.

The calculation of the dust optical properties is improved with three corrections: (1) remap the fractions of AFWA bin 1 dust in 0.2-2 μm into Mie calculation bins as Ukhov et al. (2021); (2) redistribute fractions of the dust mass based on the assumption that bin concentration is a function of natural logarithm radius as Ukhov et al. (2021); (3) increase the 8 dust size bins in Mie subroutine to 9 as 0.039-0.078, 0.078-0.156, 0.156-0.312, 0.312-0.625, 0.625-1.25, 1.25-2.5, 2.5-5.0, 5.0-10.0 μm, and 10-20 μm to distribute the AFWA bin 5 dust in 12-20 μm into bins for Mie calculation.
Reference:

Ukhov, A., Ahmadov, R., Grell, G., & Stenchikov, G. Improving dust simulations in WRF-Chem v4.1.3 coupled with the GOCART aerosol module. Geosci. Model Dev., 2021.

**Changes in Manuscript:** Please refer the description in the revised manuscript, from Page 5 L147 to Page 6 L155.

---

## Author Comment (AC2)

**Responses to the Community comment**

We would like to thank to the reviewer for giving constructive criticisms, which are very helpful in improving the quality of the manuscript. We have made major revision based on the critical comments and suggestions of the referees. The referee's comments are reproduced (*black*) along with our replies (blue) and changes made to the text (red) in the revised manuscript. All the authors have read the revised manuscript and agreed with submission in its revised form.

**Anonymous Referee #2**

**Comment NO.1:** *FR simulation has a large bias. It is known that the emission DA is sensitive to biases in the prior (e.g.: Tsikerdekis et al. 2022). I strongly recommend a calibration of the emission parameters prior to the assimilation experiments. Please give particular attention to the comments regarding the emission scheme in WRF by A. Ukhov (https://doi.org/10.5194/egusphere-2024-840-CC1)*

**Response:** Done. To reduce the underestimation of dust emission in AFWA scheme and start from a relatively unbiased simulation, the adjustable dust emission factor is calibrated and selected as 21 based on the AERONET-observed AOT and AE. This rescaling of emission can benefit the data assimilation since the Kalman filter assumes that the model is unbiased (Tsikerdekis et al., 2021). Done. In order to investigate the influence of different dust emission schemes on the dust emission during 14-23 March 2021, another experiment named FR-dustopt1 (dust_opt=1) has been conducted to compare with FR experiment (dust_opt=3). The accumulated dust emissions in Gobi desert during 14-23 March 2021 are 36.99Tg and 43.00Tg for FR and FR-dustopt1 experiments. In general, the dust emission simulated by GOCART and AFWA dust emission schemes have no significant differences in Gobi desert. Therefore, we still use AFWA dust emission scheme in the revised manuscript.

References:

Tsikerdekis, A., Schutgens, N. A. J., and Hasekamp, O. P.: Assimilating aerosol optical properties related to size and absorption from POLDER/PARASOL with an ensemble data assimilation system, Atmos. Chem. Phys., 21, 2637–2674, https://doi.org/10.5194/acp-21-2637-2021, 2021.

**Changes in Manuscript:** Please refer the description in the revised manuscript, from Page 5 Line 144 to Page 6 Line 146.

**Comment NO.2:** *The assimilated observations contain all aerosols, the simulated AOT and AE are calculated with all simulated aerosols, but only dust emissions are in the state vector. High AOD and low AE in Figure 5 indicate that the model is probably not simulating enough coarse particles (probably dust) over the sites. A simple calibration of the dust sources by a constant factor would be sufficient to improve the simulations in a similar way as the full data assimilation does here. This mismatch between the state variables and the observables is important, as it determines the ability of the data assimilation system to propagate the observational information into the model state, but it is only mentioned very superficially in the last line of the conclusions.*

**Response:** Done. To reduce the underestimation of dust emission in AFWA scheme and start from a relatively unbiased simulation, the adjustable dust emission factor is calibrated and selected as 21 based on the AERONET-observed AOT and AE. This rescaling of emission can benefit the data assimilation since the Kalman filter assumes that the model is unbiased (Tsikerdekis et al., 2021).

Done. Since the East Asian dust storms are triggered by an exceptionally strong Mongolian cyclone and accompanied by strong northwesterly wind (Gui et al., 2021), human pollutants are difficult to accumulate in the downwind areas during this period. To minimize the influences of anthropogenic emissions, only the AERONET AOT and AE dominated by dust are assimilated to optimize the dust emissions, which are chosen with AE at 440-870 nm less than 0.4 (Huneeus et al., 2011). Therefore, the observables generally match the state variables.

References:

Gui, K., Yao, W., Che, H., An, L., Zheng, Y., Li, L., Zhao, H., Zhang, L., Zhong, J., Wang, Y., and Zhang, X.: Two mega sand and dust storm events over northern China in March 2021: transport processes, historical ranking and meteorological drivers, https://doi.org/10.5194/acp-2021-933, 1 December 2021.

Huneeus, N., Schulz, M., Balkanski, Y., Griesfeller, J., Prospero, J., Kinne, S., Bauer, S., Boucher, O., Chin, M., Dentener, F., Diehl, T., Easter, R., Fillmore, D., Ghan, S., Ginoux, P., Grini, A., Horowitz, L., Koch, D., Krol, M. C., Landing, W., Liu, X., Mahowald, N., Miller, R., Morcrette, J.-J., Myhre, G., Penner, J., Perlwitz, J., Stier, P., Takemura, T., and Zender, C. S.: Global dust model intercomparison in AeroCom phase I, Atmospheric Chemistry and Physics, 11, 7781–7816, https://doi.org/10.5194/acp-11-7781-2011, 2011.

Tsikerdekis, A., Schutgens, N. A. J., and Hasekamp, O. P.: Assimilating aerosol optical properties related to size and absorption from POLDER/PARASOL with an ensemble data assimilation system, Atmos. Chem. Phys., 21, 2637–2674, https://doi.org/10.5194/acp-21-2637-2021, 2021.

**Changes in Manuscript:** Please refer the description in the revised manuscript, from Page 5 Line 144 to Page 6 Line 146 and Page 4 Line 101-102.

**Comment NO.3:** *The assimilated observations are from the Beijing area. This small spatial sample of assimilated observations, plus the way that the ensemble is constructed (by perturbing the emissions with a factor, one factor per ensemble), is likely to produce analyses emission corrections very homogeneous in the whole domain, meaning that the ratio between the analyses and first guess emissions is constant in the domain: increments of emissions in the western boundary (India and Pakistan) of Figures 1, 2 and 3 show this behaviour: the increments over Indian and Pakistan regions are similar to those of the Gobi desert, while the assimilated observations are located near Beijing. The temporal behaviour of the increments seems to show the same issue (Figure 4).*

**Response:** Dai et al. (2021) pointed that the independent emission perturbations over each model grid tend to underestimate the model spread due to the current limited ensemble members and the cancellation of neighbouring cells. Therefore, in the revised manuscript, the same random perturbation factors are used in the whole domain emission assimilation grids. The emission

correction is heterogeneous in the domain due to the perturbation of dust size distribution, the employment of horizontal localization, and the assimilation of observations from both dust source region and downwind area. As shown in Fig. 2, during 00:00-10:00 UTC 21 March 2021, the AOT DA-SZD experiment increases the dust emission in Gobi desert while it decreases the dust emission in Taklamakan desert, which indicated heterogeneous emission correction.

References:

Dai, T., Cheng, Y., Goto, D., Li, Y., Tang, X., Shi, G., and Nakajima, T.: Revealing the sulfur dioxide emission reductions in China by assimilating surface observations in WRF-Chem, Atmos. Chem. Phys., 23, 2021.

**Comment NO.4:** *The assimilation and verification period is extremely short: 4 days with very few observations (see for example Figure 16). It may be difficult to draw conclusions from such a small observational sample. A longer study period would be highly beneficial, although it is not always possible to perform.*

**Response:** Done. Although it is not always possible to perform, the assimilation and verification period are extended from 14-17 March 2021 to 14-23 March 2021 to include both a strong dust storm and a weak dust storm.

**Changes in Manuscript:** Please refer the description in the revised manuscript, Page 7 Line 194-195.

**Comment NO.5:** *I haven't found strong evidence in the manuscript showing that the assimilation of AE improves the temporal and spatial variability of emissions or AOT. Also, it would be good to explain why AE assimilation increases AOD and consequently improves the skills. The scientific reason for this is not clear in this paper. It could be because it actually introduces important size information to the system, or just because it changes the balance between the observational and background terms in the cost function (adding more weight to the observations, for example), which in turn make the increments larger, which decreases the biases and improves the scores.*

**Response:** Done.

Our results demonstrate that the additional assimilation of AE observations with consideration of the dust emission size distribution uncertainty are helpful to the optimization of dust emission through better adjustment of dust size distribution. AOT assimilation can only optimize the dust emission flux depending on the covariance between time-lagged AOT observations and the simulated total dust emission, while the additional inclusion of AE assimilation can optimize the size distribution of dust emission and the associated total flux depending on the covariance between time-lagged AE observations and the simulated dust emission in each bin.

**Changes in Manuscript:** Please refer the detailed description in Sect. 4 in the revised manuscript.

**Comment NO.6:** *I would suggest that, in order to make the study more scientifically attractive and with more enriching conclusions, the experiments should start from a relatively unbiased FR. Also, as noted by some of the co-authors in a previous article (Dai et al. 2019, last paragraph of their conclusions), the design of ensemble (with no temporal and spatial variability) and the state vector*

*(only dust emissions) or other observation operator errors can negatively affect the results. Given the strong constrain on the time and location of the assimilated observations (and their spatio-temporal representativity) these issues are, in this case, utterly important.*

**Response:** Done. To reduce the underestimation of dust emission in AFWA scheme and start from a relatively unbiased simulation, the adjustable dust emission factor is calibrated and selected as 21 based on the AERONET-observed AOT and AE. This rescaling of emission can benefit the data assimilation since the Kalman filter assumes that the model is unbiased (Tsikerdekis et al., 2021). Done. Compared with previous article, due to WRF-Chem model has uncertainty not only on dust emission but also on dust deposition (Huang et al., 2020) and dust optical properties (Di Biagio et al., 2019) in simulation, two assimilation experiments are conducted: one assimilation experiment named AOT+AE DA-SZD assimilates both AOT and AE observations with perturbation of dust size distribution, and the other experiment named AOT+AE DA assimilates both AOT and AE observations without perturbation of dust size distribution. The comparison between AOT+AE DA and AOT+AE DA-SZD experiments proves the need to considering uncertainty of dust size distribution in dust emission optimization.

References:

Di Biagio, C., Formenti, P., Balkanski, Y., Caponi, L., Cazaunau, M., Pangui, E., Journet, E., Nowak, S., Andreae, M. O., Kandler, K., Saeed, T., Piketh, S., Seibert, D., Williams, E., and Doussin, J.-F.: Complex refractive indices and single-scattering albedo of global dust aerosols in the shortwave spectrum and relationship to size and iron content, Atmos. Chem. Phys., 19, 15503–15531, https://doi.org/10.5194/acp-19-15503-2019, 2019.

Huang, Y., Kok, J. F., Kandler, K., Lindqvist, H., Nousiainen, T., Sakai, T., Adebiyi, A., and Jokinen, O.: Climate Models and Remote Sensing Retrievals Neglect Substantial Desert Dust Asphericity, Geophysical Research Letters, 47, e2019GL086592, https://doi.org/10.1029/2019GL086592, 2020.

Tsikerdekis, A., Schutgens, N. A. J., and Hasekamp, O. P.: Assimilating aerosol optical properties related to size and absorption from POLDER/PARASOL with an ensemble data assimilation system, Atmos. Chem. Phys., 21, 2637–2674, https://doi.org/10.5194/acp-21-2637-2021, 2021.

**Changes in Manuscript:** Please refer the description in the revised manuscript, from Page 5 Line 144 to Page 6 Line 146 and from Page 7 Line 194 to Page 8 Line 210.

**Comment NO.7:** *L15: Please note that AE does not resolve particle size. It is sensitive to it, but not only.*

**Response:** Done. The sentence is modified as "Ångström Exponent (AE) which, measures the wavelength-dependence of aerosol optical thickness (AOT), is significantly sensitive to large aerosol such as dust."

**Changes in Manuscript:** Please refer the modification in the revised manuscript, Page 1 Line 15-17.

**Comment NO.8:** *L17: I would suggest replacing the work "crucial" with "possible" or similar.*

**Response:** Done.

**Changes in Manuscript:** Please refer the modification in the revised manuscript, Page 1 Line 18.

**Comment NO.9:** *L21: The authors could replace the "official WRF-Chem" with the emission option they actually used.*

**Response:** The "official WRF-Chem" has been deleted.

**Comment NO.10:** *L26: Is it necessary to distinguish between Mongolia and China Gobi?*

**Response:** Agree. The Mongolia and China Gobi are merged.

**Changes in Manuscript:** Please refer to Fig. 2 in the revised manuscript.

**Comment NO.11:** *L30: Please revise the first sentence.*

**Response:** Agree. We have revised the first sentence as "Mineral dust is the most abundant atmospheric aerosol component in terms of aerosol dry mass in the atmosphere.".

**Changes in Manuscript:** Please refer the modification in the revised manuscript, Page 2 Line 33.

**Comment NO.12:** *L43-44: It is very unlikely that we could fully understand the whole dust cycle by studying just one dust storm. Also, the logic of the argument and the connections between the sentences in these two lines are not clear.*

**Response:** Done. We have rephrased the sentence: Numerical model are important tools to study severe dust storm, and the dust emission is a significant quantity to characterize dust activity. However, due to insufficient knowledge of the actual dust mechanisms, more than ten fold diversity exists in simulated East Asian dust emissions among different models (Uno et al., 2006, Kok et al., 2021), indicating dust emission is a significantly uncertain process in dust simulation.

References:

Kok, J. F., Adebiyi, A. A., Albani, S., Balkanski, Y., Checa-Garcia, R., Chin, M., et al. Contribution of the world's main dust source regions to the global cycle of desert dust (preprint). Atmospheric Chemistry and Physics. https://doi.org/10.5194/acp-2021-4, 2021.

Uno, I., Wang, Z., Chiba, M., Chun, Y. S., Gong, S. L., Hara, Y., & Jung, E. (n.d.). Dust model intercomparison (DMIP) study over Asia: Overview.

**Changes in Manuscript:** Please refer the description in the revised manuscript, Page 2 Line 44-47.

**Comment NO.13:** *L45: Please delete "simulation".*

**Response:** Done. We have deleted "simulation".

**Changes in Manuscript:** Please refer the modification in the revised manuscript, Page 2 Line 44.

**Comment NO.14:** L61: *Please add "domain" or similar: "... forecast in the global domain".*

**Response:** Accept. We add "domain".

**Changes in Manuscript:** Please refer the modification in the revised manuscript, Page 2 Line 62.

**Comment NO.15:** *L64-65: These lines are important for the manuscript, but not very precise. Please try to rewrite them.*

*L66 : Not so open. Please refer to the series of papers by Tsikerdekis et al. (ACP 2021, GMD 2022 and ACP 2023), where there is no explicit sensitivity of emissions to AE, but their usefulness can be*

*inferred.*

*L67-69: The authors just said that it was an open question, and they answer it here by saying that ground-based AE is critical for dust emissions. I think the authors are willing to say that ground-based AE is better than satellite-based AE, and can be more useful for optimizing dust emissions. Please note the accuracy of satellite retrieved AE depends on the instruments (not the same for MODIS than for multi-angle polarimeters), and on the retrievals algorithm.*

**Response:** Done. We have rewritten this paragraph: Except AOT, Ångström Exponent (AE) which measures the wavelength-dependence of AOT and is significantly sensitive to size of aerosol particle, may have a positive impact on data assimilation (Tsikerdekis et al., 2022, 2023). The estimated emission may be misrepresented by not including observations related to size (Tsikerdekis et al., 2021). However, most of the abovementioned studies have estimated new dust emissions based on the assimilation of AOT, while few studies have explored the potential benefits of aerosol size information like AE observations on improving the estimate of dust emission.

Therefore, how will the assimilation of the AE observations affects the optimization of the dust emission? It becomes an important scientific question. Due to the accuracy of satellite-retrieved AE depends on the instrument and retrieval algorithm, the ground-based AE is better than satellite-based AE and can be more useful for optimizing dust emissions.

References:

Tsikerdekis, A., Schutgens, N. A. J., and Hasekamp, O. P.: Assimilating aerosol optical properties related to size and absorption from POLDER/PARASOL with an ensemble data assimilation system, Atmos. Chem. Phys., 21, 2637–2674, https://doi.org/10.5194/acp-21-2637-2021, 2021.

Tsikerdekis, A., Schutgens, N. A. J., Fu, G., and Hasekamp, O. P.: Estimating aerosol emission from SPEXone on the NASA PACE mission using an ensemble Kalman smoother: observing system simulation experiments (OSSEs), Geosci. Model Dev., 15, 3253–3279, https://doi.org/10.5194/gmd-15-3253-2022, 2022.

Tsikerdekis, A., Hasekamp, O. P., Schutgens, N. A. J., and Zhong, Q.: Assimilation of POLDER observations to estimate aerosol emissions, Atmos. Chem. Phys., 23, 9495–9524, https://doi.org/10.5194/acp-23-9495-2023, 2023.

**Changes in Manuscript:** Please refer the description in the revised manuscript, Page 3 Line 64-71.

**Comment NO.16:** *L70: The authors say here that they use the fixed-lag Kalman smoother (and later LETKF). Can it be provided a reference to the method that was actually used? Is it Dai et al. 2019?*

**Response:** The reference Dai et al. (2019) has been provided to the method.

**Changes in Manuscript:** Please refer the modification in the revised manuscript, Page 6 Line 163.

**Comment NO.17:** *L80: No need for "direct". Or it is referring to the "direct sun" product from AERONET?*

**Response:** Accept. The "direct" has been deleted.

**Changes in Manuscript:** Please refer the modification in the revised manuscript, Page 3 Line 84.

**Comment NO.18:** *L84: This is inaccurate. It should be clearer with the formula or point to a good reference.*

**Response:** Done. AE ($\alpha$) at 440-870 nm is calculated with the AOT at 440 nm and 870 nm from the following equation: $\alpha_{440-870} = -ln(\tau_{870}/\tau_{440})/ln(870/440)$.

**Changes in Manuscript:** Please refer the modification in the revised manuscript, Page 3 Line 88-89.

**Comment NO.19:** *L86: This 0.05 seems a little arbitrary. Is there any reference for the accuracy of AERONET 440-870 AE?*

**Response:** Giles et al. (2019) pointed that the AE at 440−870 nm wavelength should not be used due to the inconsistency of the AOT spectral dependence at very low optical depth (< 0.05).
Reference:
Giles, D. M., Sinyuk, A., Sorokin, M. G., Schafer, J. S., Smirnov, A., Slutsker, I., et al. Advancements in the Aerosol Robotic Network (AERONET) Version 3 database – automated near-real-time quality control algorithm with improved cloud screening for Sun photometer aerosol optical depth (AOD) measurements. Atmospheric Measurement Techniques, 12(1), 169–209. https://doi.org/10.5194/amt-12-169-2019, 2019.

**Changes in Manuscript:** Please refer the modification in the revised manuscript, Page 3 Line 90.

**Comment NO.20:** *L90: Why 0.025? how do was estimated this value?. So far the reader does not know the spatial resolution of the WRF grid.*

**Response:** Done. The representation error has been recalculated.
Due to the representation error is related to the WRF-Chem grid resolution, the representation error in AERONET AOT and AE is calculated depending on the AOT and AE temporal variability of AERONET and WRF-Chem with 45 km horizontal resolution (Schutgens et al., 2010). By averaging results at all AERONET sites in March 2021, the relative AOT temporal variations of AERONET and WRF-Chem in 1 h interval are $0.11\tau$ and $0.1\tau$, while the AE temporal variations of AERONET and WRF-Chem in 1 h interval are 0.05 and 0.02, respectively. Therefore, the representation errors in AERONET AOT ($\tau$) and AE in the 1 h interval are $\epsilon_r = 0.01\tau$ and $\epsilon_r = 0.03$, respectively. The instrument error in AOT is defined as 0.015 and the instrument error of AE is estimated by propagating the instrument error in AOT at 440 and 870 nm (Schutgens et al., 2010).
References:
Schutgens, N. A. J., Miyoshi, T., Takemura, T., & Nakajima, T. Applying an ensemble Kalman filter to the assimilation of AERONET observations in a global aerosol transport model. Atmos. Chem. Phys., 16, 2010.

**Changes in Manuscript:** Please refer the description in the revised manuscript, from Page 3 Line 92 to Page 4 Line 100.

**Comment NO.21:** *L108: "... with 45 km ..."*

**Response:** Done. We have replaced "with the 45 km" with "with 45 km".

**Changes in Manuscript:** Please refer the modification in the revised manuscript, Page 5 Line 137.

**Comment NO.22:** *L115: size bins in diameter or radius?*

**Response:** Size bins in diameter.

**Changes in Manuscript:** Please refer the description in the revised manuscript, Page 5 Line 144.

**Comment NO.23:** *L117: MOSAIC acronym, description and need for use it is not clear here. Isn't this all done with GOCART?*

**Response:** Done. We have defined the acronym MOSAIC. The aerosol module used in this study is GOCART and the chemical composition of dust is unchanged during Mie calculation.

**Changes in Manuscript:** Please refer the description in the revised manuscript, Page 6 Line 150.

**Comment NO.24:** *L128 So the ensemble only has variability in dust emissions? Does this mean that all AOD and AE departures (AERONET minus model), where all aerosols are included, are only attributed to increases in dust emissions?*

**Response:** To investigate the influences of AERONET AE assimilation on the dust emission inversion, three assimilation experiments are conducted from 12:00 UTC on 11 March 2021 to 00:00 UTC on 24 March 2021. The assimilation experiment named AOT+AE DA assimilates both AOT and AE observations. 20 ensemble members are generated by perturbing the total dust emission, and the perturbation factor has a mean of 1 and a spread of 0.6 followed the lognormal distribution. The ensemble prediction dynamically estimates the covariance between the total dust emission and the aerosol optical properties. Due to WRF-Chem model has uncertainty not only on dust emission but also on dust deposition (Huang et al., 2020) and dust optical properties (Di Biagio et al., 2019) in simulation, two more assimilation experiments with perturbation of dust size distribution are conducted: one assimilation experiment named AOT DA-SZD only assimilates AERONET AOT observations and the other assimilation experiment named AOT+AE DA-SZD assimilates both AOT and AE observations. The ensemble prediction dynamically estimates the covariance between the dust emission in 5 dust size bins and the aerosol optical properties. 20 ensemble members are generated by perturbing the dust emission in each bin, and the perturbation factor of each bin has a mean of 1 and a spread of 0.6 followed the lognormal distribution. Correlated noise is used across the dust size bins in the perturbation for AOT DA-SZD and AOT+AE DA-SZD experiments so that noise correlation decreases with increased difference of the diameter among the bins (Di Tomaso et al., 2017). The comparison between AOT+AE DA and AOT+AE DA-SZD experiments proves the need to considering uncertainty of dust size distribution in dust emission inversion. The comparison between AOT DA-SZD and AOT+AE DA-SZD experiments proves the need of additional AE information in dust emission inversion.

Since the East Asian dust storms are triggered by an exceptionally strong Mongolian cyclone and accompanied by strong northwesterly wind (Gui et al., 2021), human pollutants are difficult to accumulate in the downwind areas during this period. To minimize the influences of anthropogenic emissions, only the AERONET AOT and AE dominated by dust are assimilated to optimize the dust emissions, which are chosen with AE at 440-870 nm less than 0.4 (Huneeus et al., 2011).

References:

Di Biagio, C., Formenti, P., Balkanski, Y., Caponi, L., Cazaunau, M., Pangui, E., Journet, E., Nowak, S., Andreae, M. O., Kandler, K., Saeed, T., Piketh, S., Seibert, D., Williams, E., and Doussin, J.-F.: Complex refractive indices and single-scattering albedo of global dust aerosols in the shortwave spectrum and relationship to size and iron content, Atmos. Chem. Phys., 19, 15503–15531, https://doi.org/10.5194/acp-19-15503-2019, 2019.

Di Tomaso, E., Schutgens, N. A. J., Jorba, O., & Pérez García-Pando, C. Assimilation of MODIS Dark Target and Deep Blue observations in the dust aerosol component of NMMB-MONARCH version 1.0. Geoscientific Model Development, 10(3), 1107–1129. https://doi.org/10.5194/gmd-10-1107-2017, 2017.

Gui, K., Yao, W., Che, H., An, L., Zheng, Y., Li, L., Zhao, H., Zhang, L., Zhong, J., Wang, Y., and Zhang, X.: Two mega sand and dust storm events over northern China in March 2021: transport processes, historical ranking and meteorological drivers, https://doi.org/10.5194/acp-2021-933, 1 December 2021.

Huang, Y., Kok, J. F., Kandler, K., Lindqvist, H., Nousiainen, T., Sakai, T., Adebiyi, A., and Jokinen, O.: Climate Models and Remote Sensing Retrievals Neglect Substantial Desert Dust Asphericity, Geophysical Research Letters, 47, e2019GL086592, https://doi.org/10.1029/2019GL086592, 2020.

Huneeus, N., Schulz, M., Balkanski, Y., Griesfeller, J., Prospero, J., Kinne, S., Bauer, S., Boucher, O., Chin, M., Dentener, F., Diehl, T., Easter, R., Fillmore, D., Ghan, S., Ginoux, P., Grini, A., Horowitz, L., Koch, D., Krol, M. C., Landing, W., Liu, X., Mahowald, N., Miller, R., Morcrette, J.-J., Myhre, G., Penner, J., Perlwitz, J., Stier, P., Takemura, T., and Zender, C. S.: Global dust model intercomparison in AeroCom phase I, Atmospheric Chemistry and Physics, 11, 7781–7816, https://doi.org/10.5194/acp-11-7781-2011, 2011.

**Changes in Manuscript:** Please refer the description in the revised manuscript, from Page 7 Line 194 to Page 8 Line 210 and Page 4 Line 101-102.

**Comment NO.25:** *L131-132: This is not clear. Please clarify.*

**Response:** Done. As shown in Fig. S1, based on the EnKS, the dust emission of WRF-Chem ensemble is optimized every 12 h, which corresponding to the assimilation time window of 12 h. Each assimilation cycle advances a time step of 12 h, and the dust emissions for 6 time steps are optimized using the observations in the 6th time step. After each assimilation cycle, the dust emission for the first time step is the final optimized result, which has been optimized 6 times and will no longer be optimized in the next cycle. The finally optimized dust emissions therefore serve as the forced dust emissions for advancing the system one time step, and they provide the initial conditions for the next assimilation cycle.

**Changes in Manuscript:** Please refer the description in the revised manuscript, Page 6 Line 164-170.

**Comment NO.26:** *L133: Please clarify if the authors are using the lagged LETKF (e.g. Schutgens*

*2012), the EnKS from Dai et al. 2019, or if it is the same method.*

**Response:** Fixed-lag LETKF and Kalman smoother are the same method from Dai et al. (2019) and Schutgens et al. (2012).

References:

Dai, Cheng, Goto, Schutgens, Kikuchi, Yoshida, et al. Inverting the East Asian Dust Emission Fluxes Using the Ensemble Kalman Smoother and Himawari-8 AODs: A Case Study with WRF-Chem v3.5.1. Atmosphere, 10(9), 543. https://doi.org/10.3390/atmos10090543, 2019.

Schutgens, N., Nakata, M., & Nakajima, T. Estimating Aerosol Emissions by Assimilating Remote Sensing Observations into a Global Transport Model. Remote Sensing, 4(11), 3528–3543. https://doi.org/10.3390/rs4113528, 2012.

**Changes in Manuscript:** Please refer the description in the revised manuscript, Page 6 Line 163-164.

**Comment NO.27:** *L137: Please revise if this is the Kalman gain. The Kalman gain in this context is clearly defined in Eq. 10 of Hunt et al. 2007, where it is the matrix in the left multiplication of the departures (in observational space), not the vector w (in ensemble space).*

**Response:** Agree. The matrix $\tilde{P}^a (Y^b)^T R^{-1}$ is called the "Kalman gain".

**Changes in Manuscript:** Please refer the description in the revised manuscript, Page 6 Line 175.

**Comment NO.28:** *L152: Again, 4D-LETKF or EnKS?*

**Response:** Fixed-lag LETKF and Kalman smoother are the same method from Dai et al. (2019) and Schutgens et al. (2012).

References:

Dai, Cheng, Goto, Schutgens, Kikuchi, Yoshida, et al. Inverting the East Asian Dust Emission Fluxes Using the Ensemble Kalman Smoother and Himawari-8 AODs: A Case Study with WRF-Chem v3.5.1. Atmosphere, 10(9), 543. https://doi.org/10.3390/atmos10090543, 2019.

Schutgens, N., Nakata, M., & Nakajima, T. Estimating Aerosol Emissions by Assimilating Remote Sensing Observations into a Global Transport Model. Remote Sensing, 4(11), 3528–3543. https://doi.org/10.3390/rs4113528, 2012.

**Changes in Manuscript:** Please refer the description in the revised manuscript, Page 6 Line 163-164.

**Comment NO.29:** *L165: Before showing the DA results, it would be useful to know exactly which and where observations were assimilated: for example, it would be sufficient to show their geographical location, time series and comparison with the FR performance in a qualitative sense. Also, if possible, the storm could be described in terms of AOD and AE from satellite observations, and if available, time-series of ground concentrations of TSP, PM20 or PM10 could useful. The later can be also useful as complement to the lidar verification in the results section.*

**Response:** Done.

As shown in Fig. 1(a), there are 5 AERONET sites with available observations from 14 March to 23 March 2021 for data assimilation, including 4 sites named as Beijing-CAMS (39.93°N,

116.32°E), Beijing (39.98°N, 116.38°E), Beijing_PKU (39.99°N, 116.31°E), and Beijing_RADI (40.00°N, 116.38°E) in the downwind area and a site near the dust source region named as Dalanzadgad (43.58°N, 104.42°E). The assimilated AOT and AE values at the AERONET sites are also given in Fig. 1(b,d). For Dalanzadgad site, the AOT values from 14 March to 17 March 2021 are significantly higher than those from 18 March to 23 March, while AE values show the opposite features.

Due to the Himawari-8 satellite observations are limited to be used during this period and the ground-based observations are difficult to be achieved, these evaluations will be shown in the future.

**Changes in Manuscript:** Please refer the description in the revised manuscript, Page 64 Line 105-111.

**Comment NO.30:** *L182-183: Interpretation of aerosol composition from a single value of AE does not seem a good practice.*

**Response:** Accept. We have deleted this sentence.

**Comment NO.31:** *Figures: If not already implemented, please try to follow the ACP recommendations on colour scales, font size, etc.*

**Response:** Done. We have tried to follow the ACP recommendations on colour scales, font size, etc.

**Comment NO.32:** *L193: It is not the best option to have different 3 figures for the 3 consecutive days.*

**Response:** Done.

**Changes in Manuscript:** Please refer the Fig. 3 and Fig. 5 in the revised manuscript.

**Comment NO.33:** *L202-203 Again, a simple plot of observations and model AOT and AE will be useful at this point (Fig 5 could also be referenced). It may just be a model bias (e.g. of other aerosols) that makes the difference.*

**Response:** Agree. Figs. 7-10 show the comparison of observed AOTs and AEs from AERONET and SONET versus the simulated ones.

To reduce the underestimation of dust emission in AFWA scheme and start from a relatively unbiased simulation, the adjustable dust emission factor is calibrated and selected as 21 based on the AERONET-observed AOT and AE. This rescaling of emission can benefit the data assimilation since the Kalman filter assumes that the model is unbiased (Tsikerdekis et al., 2021).

References:

Tsikerdekis, A., Schutgens, N. A. J., and Hasekamp, O. P.: Assimilating aerosol optical properties related to size and absorption from POLDER/PARASOL with an ensemble data assimilation system, Atmos. Chem. Phys., 21, 2637–2674, https://doi.org/10.5194/acp-21-2637-2021, 2021.

**Changes in Manuscript:** Please refer the description in the revised manuscript, Page 5 Line 144-146.

**Comment NO.34:** *L204-206: So far there is no information in the manuscript to support such a statement. This could be partly solved by showing the assimilated locations and time series*

*beforehand.*

**Response:** Done.

**Changes in Manuscript:** The assimilated locations and time series are given in Fig. 1.

**Comment NO.35:** *L213: The usual terminology indicates that it is the model or the observation operator that is inverted, not the emissions. The word "inverted" could be replaced with "posterior".*

**Response:** The word "inverted" has been replaced with "posterior".

**Changes in Manuscript:** Please refer the modification in the revised manuscript, Page 13 Line 289.

**Comment NO.36:** *L218-221: First presentation of the assimilated observations. This information needs to be presented before the results section.*

**Response:** Done. As shown in Fig. 1(a), there are 5 AERONET sites with available observations from 14 March to 23 March 2021 for data assimilation, including 4 sites named as Beijing-CAMS (39.93°N, 116.32°E), Beijing (39.98°N, 116.38°E), Beijing_PKU (39.99°N, 116.31°E), and Beijing_RADI (40.00°N, 116.38°E) in the downwind area and a site near the dust source region named as Dalanzadgad (43.58°N, 104.42°E). The assimilated AOT and AE values at the AERONET sites are also given in Fig. 1(b,d). For Dalanzadgad site, the AOT values from 14 March to 17 March 2021 are significantly higher than those from 18 March to 23 March, while AE values show the opposite features.

**Changes in Manuscript:** Please refer the description in the revised manuscript, Page 64 Line 105-111.

**Comment NO.37:** *L223: Probably the word "obviously" is not really needed.*

**Response:** This sentence has been deleted.

**Comment NO38:** *L228-230: Evaluation scores are computed by mixing assimilated and non-assimilated observations rather than separately, making the evaluations difficult to interpret.*

**Response:** Assimilated and non-assimilated observations have been separated.

**Changes in Manuscript:** Please refer to Fig. 7 and Fig. 8 in the revised manuscript.

**Comment NO.39:** *L233-234: The authors claim: "... and thus capture the aerosol spatiotemporal variation characteristics during dust transportation". True or not, this claim is not fully supported by the above results. Please revise this statement, as it is not obvious that the DA is doing more than just scaling the dust burden.*

**Response:** Done. This wrong statement has been deleted.

In this study, to reduce the underestimation of dust emission in AFWA scheme and start from a relatively unbiased simulation, the adjustable dust emission factor is calibrated and selected as 21 based on the AERONET-observed AOT and AE. This rescaling of emission can benefit the data assimilation since the Kalman filter assumes that the model is unbiased (Tsikerdekis et al., 2021).

To investigate the influences of AERONET AE assimilation on the dust emission inversion, three assimilation experiments are conducted from 12:00 UTC on 11 March 2021 to 00:00 UTC on 24 March 2021. The assimilation experiment named AOT+AE DA assimilates both AOT and AE

observations. 20 ensemble members are generated by perturbing the total dust emission, and the perturbation factor has a mean of 1 and a spread of 0.6 followed the lognormal distribution. The ensemble prediction dynamically estimates the covariance between the total dust emission and the aerosol optical properties. Due to WRF-Chem model has uncertainty not only on dust emission but also on dust deposition (Huang et al., 2020) and dust optical properties (Di Biagio et al., 2019) in simulation, two more assimilation experiments with perturbation of dust size distribution are conducted: one assimilation experiment named AOT DA-SZD only assimilates AERONET AOT observations and the other assimilation experiment named AOT+AE DA-SZD assimilates both AOT and AE observations. The ensemble prediction dynamically estimates the covariance between the dust emission in 5 dust size bins and the aerosol optical properties. 20 ensemble members are generated by perturbing the dust emission in each bin, and the perturbation factor of each bin has a mean of 1 and a spread of 0.6 followed the lognormal distribution. Correlated noise is used across the dust size bins in the perturbation for AOT DA-SZD and AOT+AE DA-SZD experiments so that noise correlation decreases with increased difference of the diameter among the bins (Di Tomaso et al., 2017). The comparison between AOT+AE DA and AOT+AE DA-SZD experiments proves the need to considering uncertainty of dust size distribution in dust emission inversion. The comparison between AOT DA-SZD and AOT+AE DA-SZD experiments proves the need of additional AE information in dust emission inversion.

References:

Di Biagio, C., Formenti, P., Balkanski, Y., Caponi, L., Cazaunau, M., Pangui, E., Journet, E., Nowak, S., Andreae, M. O., Kandler, K., Saeed, T., Piketh, S., Seibert, D., Williams, E., and Doussin, J.-F.: Complex refractive indices and single-scattering albedo of global dust aerosols in the shortwave spectrum and relationship to size and iron content, Atmos. Chem. Phys., 19, 15503–15531, https://doi.org/10.5194/acp-19-15503-2019, 2019.

Di Tomaso, E., Schutgens, N. A. J., Jorba, O., & Pérez García-Pando, C. Assimilation of MODIS Dark Target and Deep Blue observations in the dust aerosol component of NMMB-MONARCH version 1.0. Geoscientific Model Development, 10(3), 1107–1129. https://doi.org/10.5194/gmd-10-1107-2017, 2017.

Huang, Y., Kok, J. F., Kandler, K., Lindqvist, H., Nousiainen, T., Sakai, T., Adebiyi, A., and Jokinen, O.: Climate Models and Remote Sensing Retrievals Neglect Substantial Desert Dust Asphericity, Geophysical Research Letters, 47, e2019GL086592, https://doi.org/10.1029/2019GL086592, 2020.

Tsikerdekis, A., Schutgens, N. A. J., and Hasekamp, O. P.: Assimilating aerosol optical properties related to size and absorption from POLDER/PARASOL with an ensemble data assimilation system, Atmos. Chem. Phys., 21, 2637–2674, https://doi.org/10.5194/acp-21-2637-2021, 2021.

**Changes in Manuscript:** Please refer the description in the revised manuscript, Page 5 Line 144-146 and from Page 7 Line 194 to Page 8 Line 21.

**Comment NO.40:** *Figure 5: Please define this MFE. The standard definition is in the range [0,2].*

**Response:** Done. MFE has been defined and recalculated.

**Changes in Manuscript:** Please refer the description in the revised manuscript, Page 13 Line 291-294.

**Comment NO.41:** *L248-258: Differences between model experiments are clear, but it is extremely difficult to draw conclusions about model skill with such localised and small numbers of data points.*

**Response:** The assimilation and verification period are extended from 14-17 March 2021 to 14-23 March 2021 to include a strong dust storm and a weak dust storm, therefore, the data points are enough for drawing conclusions about model skill.

**Comment NO.42:** *L258: Again, AE could be useful to improve emissions, but it is not required.*

**Response:** Our results demonstrate that the additional assimilation of AE observations with consideration of the dust emission size distribution uncertainty are helpful to the optimization of dust emission through better adjustment of dust size distribution. AOT assimilation can only optimize the dust emission flux depending on the covariance between time-lagged AOT observations and the simulated total dust emission, while the additional inclusion of AE assimilation can optimize the size distribution of dust emission and the associated total flux depending on the covariance between time-lagged AE observations and the simulated dust emission in each bin.

**Changes in Manuscript:** Please refer the detailed description in Sect. 4 in the revised manuscript.

**Comment NO.43:** *L280 and 282: Please replace AOD+AR by AOD+AE. Please follow the ACP guidelines for figure composition.*

**Response:** Done. We have replaced "AOD+AR" with "AOD+AE". We have followed the ACP guidelines for figure composition.

**Comment NO.44:** *L319: Please rewrite "in better consistent".*

**Response:** "in better consistent" has been deleted.

**Comment NO.45:** *L324: "Innovative superiority" may be an overstatement.*

**Response:** "Innovative superiority" has been deleted.

---

## Author Comment (AC3)

**Responses to the Community comment**

We would like to thank to the reviewer for giving constructive criticisms, which are very helpful in improving the quality of the manuscript. We have made major revision based on the critical comments and suggestions of the referees. The referee's comments are reproduced (*black*) along with our replies (blue) and changes made to the text (red) in the revised manuscript. All the authors have read the revised manuscript and agreed with submission in its revised form.

**Anonymous Referee #1**

**Comment NO.1:** *The manuscript, titled "Improving estimation of a record breaking East Asian dust storm emission with lagged aerosol Ångström Exponent observations", by Yueming Cheng et al. is a data assimilation (DA) case study for improving dust storm predictions in China using WRF-Chem. The authors chose a dust storm in Mar 2021 as a case study and successfully showed that using AERONET observations for DA could improve hindcasting ability of dust in WRF-Chem. Authors further showed that employing the Angstrom Exponent (AE) benefits more than employing the aerosol optical thickness (AOT) in improving DA results. The methodology is generally scientifically valid and clear in presentation, although there are occasional grammar issues or unclear descriptions that could use a little more editing. The main issue I thought was that there are insufficient science discussions to the results, such as why AE benefits more than AOT, or how could the WRF-Chem dust model be improved. The manuscript so far feels a little more like a technical report on improving hindcasts rather than a scientific development on our understanding of dust modelling. I have a few major comments below for science, and I have some other specific comments on technical questions or presentation. I suggest major revision for the current revision.*

**Response:** Done.The occasional grammar issues or unclear descriptions have been rewritten.

Dust emission is a significantly uncertain process in dust simulation. In AFWA module, The dust emission factor is an important parameter needed tuning. Due to manual tuning is computationally expensive, it is beneficial to replace the tuning process with an automatic parameter estimation system to improve dust emission simulation. In this study, data assimilation, which feeds the observation information into numerical model, can be a valuable tool to automatic adjust dust emission parameters for the optimization of the estimates of dust emissions. Aerosol optical thickness (AOT) represents the total amount of atmospheric column, however, lacks the aerosol size information. Ångström Exponent (AE) which measures the wavelength-dependence of AOT and is significantly sensitive to size of aerosol particle, may have a positive impact on data assimilation (Tsikerdekis et al., 2022, 2023).

Due to WRF-Chem model has uncertainties not only on dust emission but also on dust deposition (Huang et al., 2020) and dust optical properties (Di Biagio et al., 2019) in simulation, two assimilation experiments with perturbation of dust emission and size distribution are conducted. One assimilation experiment named AOT DA-SZD only assimilates AERONET AOT observations, and the other assimilation experiment named AOT+AE DA-SZD assimilates both AERONET AOT

and AE observations. The comparison between AOT DA-SZD and AOT+AE DA-SZD experiments shows the effects of the additional AE information on dust emission optimization. Our results demonstrate that the additional assimilation of AE observations with consideration of the dust emission size distribution uncertainty are helpful to the optimization of dust emission through better adjustment of dust size distribution. AOT assimilation can only optimize the dust emission flux depending on the covariance between time-lagged AOT observations and the simulated total dust emission, while the additional inclusion of AE assimilation can optimize the size distribution of dust emission and the associated total flux depending on the covariance between time-lagged AE observations and the simulated dust emission in each bin. The sufficient science discussions to the results are given in Sect. 3.

References:

Di Biagio, C., Formenti, P., Balkanski, Y., Caponi, L., Cazaunau, M., Pangui, E., Journet, E., Nowak, S., Andreae, M. O., Kandler, K., Saeed, T., Piketh, S., Seibert, D., Williams, E., and Doussin, J.-F.: Complex refractive indices and single-scattering albedo of global dust aerosols in the shortwave spectrum and relationship to size and iron content, Atmos. Chem. Phys., 19, 15503–15531, https://doi.org/10.5194/acp-19-15503-2019, 2019.

Huang, Y., Kok, J. F., Kandler, K., Lindqvist, H., Nousiainen, T., Sakai, T., Adebiyi, A., and Jokinen, O.: Climate Models and Remote Sensing Retrievals Neglect Substantial Desert Dust Asphericity, Geophysical Research Letters, 47, e2019GL086592, https://doi.org/10.1029/2019GL086592, 2020.

Tsikerdekis, A., Schutgens, N. A. J., Fu, G., and Hasekamp, O. P.: Estimating aerosol emission from SPEXone on the NASA PACE mission using an ensemble Kalman smoother: observing system simulation experiments (OSSEs), Geosci. Model Dev., 15, 3253–3279, https://doi.org/10.5194/gmd-15-3253-2022, 2022.

Tsikerdekis, A., Hasekamp, O. P., Schutgens, N. A. J., and Zhong, Q.: Assimilation of POLDER observations to estimate aerosol emissions, Atmos. Chem. Phys., 23, 9495–9524, https://doi.org/10.5194/acp-23-9495-2023, 2023.

**Changes in Manuscript:** Please refer the description in the revised manuscript, Page 2 Line 44-47, Page 3 Line 64-65, Page 7 Line 194-208, and Sect. 3.

**Comment NO.2:** *I think there needs to be some science discussions on what caused the biases in WRF Chem dust emissions in China. The paper currently leaves readers with puzzles regarding why WRF-Chem underestimates dust so much. Is it problems in simulated dust emissions, lifetime (dust deposition), or optical properties? If the bias comes from emission, is it a problem in wind speed, soil moisture, vegetation, or other met fields? If it's deposition or optics, is there anything to do with size distribution, dust particle shape, or dust refractive index? Does the dust underestimation also occur over the rest of the world in WRF-Chem? How does changing the a priori emissions (using other dust_opt or other emission schemes) alter the FR underestimations? Modellers would like to know how could our process-based dust understanding benefit from the*

*insights from this DA study.*

**Response:** Done. Due to the predicted meteorological fields such as wind and soil moisture are constrained by NCEP Final (FNL) analysis, dust emission parameterization is a significantly uncertain process in the dust emission simulation. As Su and Fung. (2015) pointed out the underestimation of the dust emission over the Gobi desert by the AFWA scheme, in this study, to reduce the underestimation of dust emission in AFWA scheme and start from a relatively unbiased simulation, the adjustable dust emission factor is calibrated and selected as 21 based on the AERONET-observed AOT and AE. After the bias calibration of dust emission, due to WRF-Chem model has uncertainties not only on dust emission but also on dust deposition (Huang et al., 2020) and dust optical properties (Di Biagio et al., 2019) in simulation, two assimilation experiments with perturbation of dust emission and their size distribution are conducted. One assimilation experiment named AOT DA-SZD only assimilates AERONET AOT observations, and the other assimilation experiment named AOT+AE DA-SZD assimilates both AERONET AOT and AE observations. The comparison between AOT DA-SZD and AOT+AE DA-SZD experiments shows the effects of the additional AE information on dust emission optimization.

References:

Di Biagio, C., Formenti, P., Balkanski, Y., Caponi, L., Cazaunau, M., Pangui, E., Journet, E., Nowak, S., Andreae, M. O., Kandler, K., Saeed, T., Piketh, S., Seibert, D., Williams, E., and Doussin, J.-F.: Complex refractive indices and single-scattering albedo of global dust aerosols in the shortwave spectrum and relationship to size and iron content, Atmos. Chem. Phys., 19, 15503–15531, https://doi.org/10.5194/acp-19-15503-2019, 2019.

Huang, Y., Kok, J. F., Kandler, K., Lindqvist, H., Nousiainen, T., Sakai, T., Adebiyi, A., and Jokinen, O.: Climate Models and Remote Sensing Retrievals Neglect Substantial Desert Dust Asphericity, Geophysical Research Letters, 47, e2019GL086592, https://doi.org/10.1029/2019GL086592, 2020.

Su, L. and Fung, J. C. H.: Sensitivities of WRF-Chem to dust emission schemes and land surface properties in simulating dust cycles during springtime over East Asia, JGR Atmospheres, 120, https://doi.org/10.1002/2015JD023446, 2015.

**Changes in Manuscript:** Please refer the description in the revised manuscript, from Page 5 Line 144 to Page 6 Line 146 and Page 7 Line 194-208.

**Comment NO.3:** *It looks like authors attributed all the differences/biases between AERONET-measured and WRF-simulated AOT to dust. Could the biases be attributed to other natural and anthropogenic emissions? Although you only assimilated three days where dust was dominant, there must be some strong anthropogenic and natural emissions that got captured by AERONET, especially over Beijing, a heavily polluted metropolitan area. You only used AE values < 0.4 for evaluation to focus on dust, but it seems you didn't do the same when doing the DA. From my point of view, it could be better to use the coarse-mode AERONET AOT from the spectral deconvolution algorithm (SDA) to do the DA since all fine-mode aerosols are truncated and only dust/sea-salt*

*remains.*

**Response:** Done. Since the East Asian dust storms are triggered by an exceptionally strong Mongolian cyclone and accompanied by strong northwesterly wind (Gui et al., 2021), human pollutants are difficult to accumulate in the downwind areas during this period. To minimize the influences of anthropogenic emissions, only the AERONET AOT and AE dominated by dust are assimilated to optimize the dust emissions, which are chosen with AE at 440-870 nm less than 0.4 (Huneeus et al., 2011). Therefore, the observables generally match the state variables.

Thank you for your advice. The coarse-mode AERONET AOT from the spectral deconvolution algorithm (SDA) is useful for data assimilation. However, due to the dust particle from 0.2 to 2 μm in diameter simulated by WRF-Chem is partly included in the fine-mode fraction of SDA retrievals (O'Neill et al., 2001, 2023), it is difficult to construct new observation operator and this important work will be completed in the future.

References:

Gui, K., Yao, W., Che, H., An, L., Zheng, Y., Li, L., Zhao, H., Zhang, L., Zhong, J., Wang, Y., and Zhang, X.: Two mega sand and dust storm events over northern China in March 2021: transport processes, historical ranking and meteorological drivers, https://doi.org/10.5194/acp-2021-933, 1 December 2021.

Huneeus, N., Schulz, M., Balkanski, Y., Griesfeller, J., Prospero, J., Kinne, S., Bauer, S., Boucher, O., Chin, M., Dentener, F., Diehl, T., Easter, R., Fillmore, D., Ghan, S., Ginoux, P., Grini, A., Horowitz, L., Koch, D., Krol, M. C., Landing, W., Liu, X., Mahowald, N., Miller, R., Morcrette, J.-J., Myhre, G., Penner, J., Perlwitz, J., Stier, P., Takemura, T., and Zender, C. S.: Global dust model intercomparison in AeroCom phase I, Atmospheric Chemistry and Physics, 11, 7781–7816, https://doi.org/10.5194/acp-11-7781-2011, 2011.

O'Neill, N. T., Eck, T. F., Holben, B. N., Smirnov, A., Dubovik, O., and Royer, A.: Bimodal size distribution influences on the variation of Angstrom derivatives in spectral and optical depth space, J. Geophys. Res., 106, 9787–9806, https://doi.org/10.1029/2000JD900245, 2001.

O'Neill, N. T., Ranjbar, K., Ivănescu, L., Eck, T. F., Reid, J. S., Giles, D. M., Pérez-Ramírez, D., and Chaubey, J. P.: Relationship between the sub-micron fraction (SMF) and fine-mode fraction (FMF) in the context of AERONET retrievals, Atmos. Meas. Tech., 16, 1103–1120, https://doi.org/10.5194/amt-16-1103-2023, 2023.

**Changes in Manuscript:** Please refer the description in the revised manuscript, Page 4 Line 101-102 and Page 20 Line 403-404.

**Comment NO.4:** *Even if all biases in AOT/AE were assumed to be due to dust, currently the authors attribute all differences/biases between observations and simulations to emissions and only correct emissions. This presumes there are no biases in dust settling/deposition and dust optical properties. But if so, assimilating AOTs should correct most of the error, and AE would not be needed. A science discussion is needed on why the AE information could further reduce bias. Authors could not just conclude that the more you use for DA the better the hindcast results. My thought is that if AE is*

*additionally needed for DA, this either means there are problems in AERONET AOT, or (more likely) WRF-Chem has problems not only on dust emissions but also on dust optics. Studies also have pointed out issues in both settling velocity (e.g., Huang et al., 2020) and optics (e.g., Di Biagio et al., 2019) in models. If so, it does not make so much sense to attribute all AOT/AE biases to emissions to compensate other errors in WRF-Chem. Maybe optics should also be inverted, not just emissions.*

**Response:** Done. To investigate the influences of AERONET AOT and AE assimilation on the dust emission optimization, three assimilation experiments are conducted from 12:00 UTC on 11 March 2021 to 00:00 UTC on 24 March 2021. Due to WRF-Chem model has uncertainties not only on dust emission but also on dust deposition (Huang et al., 2020) and dust optical properties (Di Biagio et al., 2019) in simulation, two assimilation experiments with perturbation of dust emission and size distribution are conducted. One assimilation experiment named AOT DA-SZD only assimilates AERONET AOT observations, and the other assimilation experiment named AOT+AE DA-SZD assimilates both AERONET AOT and AE observations. 20 ensemble members are generated by perturbing the dust emission in each bin, and the perturbation factor of each bin has a mean of 1 and a spread of 0.6 followed the lognormal distribution. Correlated noise is used across the dust size bins in the perturbation, and the noise correlation decreases with increased difference of the diameter among the bins (Di Tomaso et al., 2017). The ensemble prediction dynamically estimates the covariance between the dust emission in each bin and the aerosol optical properties. The comparison between AOT DA-SZD and AOT+AE DA-SZD experiments shows the effects of the additional AE information on dust emission optimization. The effects of dust emission size distribution perturbation are investigated by one additional assimilation experiment named AOT+AE DA, which is conducted as same as the AOT+AE DA-SZD experiment except the 20 ensemble members are generated by perturbing the dust emission in each bin with same perturbation factor.

References:

Di Biagio, C., Formenti, P., Balkanski, Y., Caponi, L., Cazaunau, M., Pangui, E., Journet, E., Nowak, S., Andreae, M. O., Kandler, K., Saeed, T., Piketh, S., Seibert, D., Williams, E., and Doussin, J.-F.: Complex refractive indices and single-scattering albedo of global dust aerosols in the shortwave spectrum and relationship to size and iron content, Atmos. Chem. Phys., 19, 15503–15531, https://doi.org/10.5194/acp-19-15503-2019, 2019.

Di Tomaso, E., Schutgens, N. A. J., Jorba, O., & Pérez García-Pando, C. Assimilation of MODIS Dark Target and Deep Blue observations in the dust aerosol component of NMMB-MONARCH version 1.0. Geoscientific Model Development, 10(3), 1107–1129. https://doi.org/10.5194/gmd-10-1107-2017, 2017.

Huang, Y., Kok, J. F., Kandler, K., Lindqvist, H., Nousiainen, T., Sakai, T., Adebiyi, A., and Jokinen, O.: Climate Models and Remote Sensing Retrievals Neglect Substantial Desert Dust Asphericity, Geophysical Research Letters, 47, e2019GL086592,

https://doi.org/10.1029/2019GL086592, 2020.

**Changes in Manuscript:** Please refer the description in the revised manuscript, Page 7 Line 194-208.

**Comment NO.5:** *Line 78: It's a little difficult to grasp how many AERONET stations you used for assimilation. Please state in text. I suggest plotting out the locations of the AERONET stations, with values of AOT and AE, either in Fig. 1 or in SI.*

**Response:** Done. As shown in Fig. 1(a), there are 5 AERONET sites with available observations from 14 March to 23 March 2021 for data assimilation, including 4 sites named as Beijing-CAMS (39.93°N, 116.32°E), Beijing (39.98°N, 116.38°E), Beijing_PKU (39.99°N, 116.31°E), and Beijing_RADI (40.00°N, 116.38°E) in the downwind area and a site near the dust source region named as Dalanzadgad (43.58°N, 104.42°E). The assimilated AOT and AE values at the AERONET sites are also given in Fig. 1(b,d). For Dalanzadgad site, the AOT values from 14 March to 17 March 2021 are significantly higher than those from 18 March to 23 March, while AE values show the opposite features.

**Changes in Manuscript:** The AERONET stations used for assimilation are stated in Page 4 Line 105-111 and the locations of the AERONET stations are plotted in Fig. 1.

**Comment NO.6:** *Lines 87-88: Authors wrote two observation errors: "observation error is a sum of representation error and observation error", which is confusing. Maybe use another word like instrument error for the latter one?*

**Response:** Done. We have replaced the latter "observation error" with "instrument error".

**Changes in Manuscript:** Please refer the modification in the revised manuscript, Page 3 Line 92.

**Comment NO.7:** *Line 89: I glanced through Schmid 1999 but didn't see such characterization of representation error. Schmid 99 was not a WRF-Chem modelling study either. Why should representation error be 0.055τ? Should it be more related to WRF-Chem grid resolution? Also, please define τ.*

**Response:** Done. The representation error has been recalculated.

Due to the representation error is related to the WRF-Chem grid resolution, the representation error in AERONET AOT and AE is calculated depending on the AOT and AE temporal variability of AERONET and WRF-Chem with 45 km horizontal resolution (Schutgens et al., 2010). By averaging results at all AERONET sites in March 2021, the relative AOT temporal variations of AERONET and WRF-Chem in 1 h interval are $0.11\tau$ and $0.1\tau$, while the AE temporal variations of AERONET and WRF-Chem in 1 h interval are 0.05 and 0.02, respectively. Therefore, the representation errors in AERONET AOT ($\tau$) and AE in the 1 h interval are $\epsilon_r = 0.01\tau$ and $\epsilon_r = 0.03$, respectively. The instrument error in AOT is defined as 0.015 and the instrument error of AE is estimated by propagating the instrument error in AOT at 440 and 870 nm (Schutgens et al., 2010).
References:

Schutgens, N. A. J., Miyoshi, T., Takemura, T., & Nakajima, T. Applying an ensemble Kalman filter to the assimilation of AERONET observations in a global aerosol transport model. Atmos.

Chem. Phys., 16, 2010.

**Changes in Manuscript:** Please refer the description in the revised manuscript, from Page 3 Line 92 to Page 4 Line 100.

**Comment NO.8:** *Line 92: I think it is needed to state why authors chose to assimilate AERONET instead of SONET or CALIOP. It looks random to me.*

**Response:** AErosol RObotic NETwork (AERONET), which both include ground-based AOT and AE observations, is chosen as the assimilated observations to investigate the sensitivity of dust emission to size information in this study. The Sun-Sky Radiometer Observation Network (SONET) and CALIPSO observations are used for independent validation.

**Changes in Manuscript:** Please refer the description in the revised manuscript, Page 3 Line 72-73 and Line 76-77.

**Comment NO.9:** *Line 93: Please also plot out the locations of the SONET sites and their values of AOT and AE, in main text or supplement.*

**Response:** Done. As shown in Fig. 1(a), there are 5 SONET sites with available observations from 14 March to 23 March 2021, including: Yanqihu (40.40°N, 116.67°E), Beijing (40.00°N, 116.37°E), Jiaozuo (35.18°N, 113.20°E), Songshan (34.53°N, 113.09°E), and Zhengzhou (34.70°N, 113.66°E). The AOT and AE values at the SONET sites are also given in Fig. 1 (c,e). Similar with Dalanzadgad site, Jiaozuo, Songshan, and Zhengzhou sites experience a stronger dust process from 14 March to 17 March 2021 with higher AOTs and lower AEs.

**Changes in Manuscript:** The locations of the SONET sites and their values of AOT and AE are plotted in Fig. 1.

**Comment NO.10:** *Line 117: Please define the acronym MOSAIC. I am not sure how important this modification is if you don't concern dust chemistry, since MOSAIC never appeared in the text again. How are the spatiotemporal distributions of metal ions changed through this modification?*

**Response:** Done. We have defined the acronym MOSAIC. The aerosol module used in this study is GOCART and the chemical composition of dust is unchanged during Mie calculation.

**Changes in Manuscript:** Please refer the description in the revised manuscript, Page 6 Line 150.

**Comment NO.11:** *Line 140: Again, it looks like the whole difference between AERONET AOTs and simulated AOTs are attributed toward dust emission biases. Can't there be biases from other natural and anthropogenic emissions?*

**Response:** Since the East Asian dust storms are triggered by an exceptionally strong Mongolian cyclone and accompanied by strong northwesterly wind (Gui et al., 2021), human pollutants are difficult to accumulate in the downwind areas during this period. To minimize the influences of anthropogenic emissions, only the AERONET AOT and AE dominated by dust are assimilated to optimize the dust emissions, which are chosen with AE at 440-870 nm less than 0.4 (Huneeus et al., 2011). Therefore, the whole difference between AERONET AOTs and simulated AOTs are attributed toward dust emission biases.

References:

Gui, K., Yao, W., Che, H., An, L., Zheng, Y., Li, L., Zhao, H., Zhang, L., Zhong, J., Wang, Y., and Zhang, X.: Two mega sand and dust storm events over northern China in March 2021: transport processes, historical ranking and meteorological drivers, https://doi.org/10.5194/acp-2021-933, 1 December 2021.

Huneeus, N., Schulz, M., Balkanski, Y., Griesfeller, J., Prospero, J., Kinne, S., Bauer, S., Boucher, O., Chin, M., Dentener, F., Diehl, T., Easter, R., Fillmore, D., Ghan, S., Ginoux, P., Grini, A., Horowitz, L., Koch, D., Krol, M. C., Landing, W., Liu, X., Mahowald, N., Miller, R., Morcrette, J.-J., Myhre, G., Penner, J., Perlwitz, J., Stier, P., Takemura, T., and Zender, C. S.: Global dust model intercomparison in AeroCom phase I, Atmospheric Chemistry and Physics, 11, 7781–7816, https://doi.org/10.5194/acp-11-7781-2011, 2011.

**Comment NO.12:** *Line 145: I am not sure if the error covariance includes forward model error (that is, error in the WRF-Chem H operator). Please clarify.*

**Response:** The analysis error covariance includes forward model error. WRF-Chem is served as observation operator and the dust emissions in each size bin are perturbed independently for AOT DA-SZD and AOT+AE DA-SZD experiments. Perturbation in the dust size distribution can lead to variations in the deposition process and optical calculations, further resulting in modifications to the meteorological field due to aerosol-radiation interactions. Therefore, forward model error is included in the error covariance.

**Comment NO.13:** *Line 155: A localization length of 600 km sounds a little too big to me for assimilating AERONET data. It is almost a meso-synoptic length scale and is much bigger than your WRF-Chem horizontal grid resolution.*

**Response:** The localization length of 600 km, which obtained through tuning, is selected as a reasonable parameter for the dust emission inversion used in Dai et al. (2019). Asian dust can be transported over long distances and affect areas far downwind. The localization length determines the assimilated observations in the horizontal space. Localization length larger than 600 km causes more observations be assimilated for the analyses at a grid point, while localization length less than 600 km makes it difficult for the time-lagged observations in downwind areas to be utilized.

References:

Dai, Cheng, Goto, Schutgens, Kikuchi, Yoshida, et al. Inverting the East Asian Dust Emission Fluxes Using the Ensemble Kalman Smoother and Himawari-8 AODs: A Case Study with WRF-Chem v3.5.1. Atmosphere, 10(9), 543. https://doi.org/10.3390/atmos10090543, 2019.

**Comment NO.13:** *Line 155: "Grid centroid" instead of "centre grid"?*

**Response:** Accept. We have replaced "centre grid" with "grid centroid".

**Changes in Manuscript:** Please refer the modification in the revised manuscript, Page 7 Line 191.

**Comment NO.14:** *Figure 1: It's interesting that using AE measurements in Beijing could lead to changes in the posterior AE (AOT+AE DA) over Taklimakan, or even in India, in comparison to the a priori AE (FR). How does DA generate emission changes in Taklimakan and India, if you were using AERONET sites in Beijing?*

**Response:** In the original manuscript, all the AERONET sites in the domain are used for data assimilation, hence, the posterior AEs in Taklimakan and India are both changed in comparison to the a priori AEs. In the revised manuscript, only 5 AERONET sites in Fig. 1 are used for data assimilation, hence, the regions (e.g., India) beyond the 600 km observation localization length are generally unchanged. If we only using AERONET sites in Beijing, DA is difficult to generate obviously emission changes in Taklimakan and India because of the 600 km localization length.

**Comment NO.15:** *Line 203-206: It is not clear what this means. There were no observations because AERONET sites were down, like because of the dust storm? Please rephrase. Does this mean if you use the observations on 14-15 March, dust emissions would be even higher? Please clarify.*

**Response:** We have deleted the sentence. It doesn't mean if we use the observations on 14-15 March, dust emissions would be even higher. Based on the assumption that the model background error covariance is correct and reasonable, the optimization of dust emission is generally invariant with the assimilated observations at different times.

**Comment NO.16:** *Line 208: I suggest adding map plots on the prior error of the WRF FR emissions, as well as the posterior errors of inverted emissions for the AOT DA and the AOT+AE DA cases. It helps visualize how adding AOT and AE for DA reduces the posterior errors of DA emissions.*

**Response:** Accept. The ratios between posterior error of dust emission and the prior one in each dust bin for the three assimilation experiments during 14-17 March 2021 and 18-23 March 2021 are given in Fig. 4 and Fig. 6.

**Changes in Manuscript:** Please refer the detailed description in Sect. 4.1 in the revised manuscript.

**Comment NO.17:** *Figure 4: When you say "aggregate", is this plot averaged across or summed across the domain? Please clarify in text.*

**Response:** The plot is summed across the domain.

**Changes in Manuscript:** Please refer the caption of Fig. 2 in the revised manuscript.

**Comment NO.18:** *Line 213: I suggest either saying posterior and prior dust emissions, or a posteriori and a priori dust emissions.*

**Response:** Done.

**Changes in Manuscript:** Please refer the modification in the revised manuscript.

**Comment NO.19:** *Lines 217-219: So, did you use more AERONET sites than listed here for DA above? Or are these all the sites used for DA? I am still confused, please clarify.*

**Response:** In the revised manuscript, the AERONET sites used for data assimilation are same as those for self-validation.

**Changes in Manuscript:** Please refer the detailed description of assimilated AERONET observations in the revised manuscript, Page 4 Line 105-111.

**Comment NO.20:** *Line 236: Here authors should suggest scientific reasons for why using AE would benefit DA so much, while AOT DA would not.*

**Response:** Done.

Our results demonstrate that the additional assimilation of AE observations with consideration of the dust emission size distribution uncertainty are helpful to the optimization of dust emission through better adjustment of dust size distribution. AOT assimilation can only optimize the dust emission flux depending on the covariance between time-lagged AOT observations and the simulated total dust emission, while the additional inclusion of AE assimilation can optimize the size distribution of dust emission and the associated total flux depending on the covariance between time-lagged AE observations and the simulated dust emission in each bin.

**Changes in Manuscript:** Please refer the detailed description in Sect. 4 in the revised manuscript.

**Comment NO.21:** *Line 248: What's the reason of selecting these two SONET sites but not the other two? It looks a little random here. I suggest plotting the comparisons for Songshan and Jiaozuo in main text or supplement too.*

**Response:** Done.

**Changes in Manuscript:** The comparison of the simulated AOTs and AEs with the observed ones over all the AERONET and SONET sites are given in Fig. 9 and Fig. 10. Please refer the detailed description in Sect. 4.2 in the revised manuscript.

**Comment NO.22:** *Line 255: From here on, I start to find the message for the next few subsections a bit repetitive, stating that the AOT+AE DA run is better than the FR run and the AOT DA run. The manuscript could use a little rewriting to make the discussion and message more succinct.*

**Response:** Done.

**Changes in Manuscript:** Please refer the detailed description in Sect. 4 in the revised manuscript.

**Comment NO.23:** *Figure 6: It seems to readers there are insufficient SONET data points for the time series plot. I suggest authors also include SONET data points for AE > 0.4 since you used it for DA, in any color other than grey.*

**Response:** As shown in Fig. 9 and Fig. 10, we extended the experiment period from 14-17 March 2021 to 14-23 March 2021 to get enough SONET data points. Due to only the AERONET observations with AE<0.4 are used for data assimilation, it is not necessary to include SONET data points for AE > 0.4.

**Comment NO.24:** *Figure 7-9: These are nice plots. Though, readers find the message across Figs. 7-9 a little repetitive. I would suggest showing the extinction coefficient (second rows) and skip the AOT plots (first rows), and maybe combine second rows of all three figures together. The first rows could be put in supplement.*

**Response:** Done.

**Changes in Manuscript:** Please refer the extinction coefficient plots in Fig. 12 in the revised manuscript and the AOT plots in Fig. S8 in supplement.

---

## Editor Decision (ED1)

Thanks for spending a lot of effort revising the manuscript following the comments and suggestions. I think most of my comments and suggestions are addressed. I just wanted to echo what the 2nd reviewer has said, that the DA gave a relatively homogeneous dust emission change across the domain (Fig. 3a and 3b-d). This makes sense because the AERONET sites are far away from the two deserts. I think partly due to this reason, in the introduction and conclusion sections the authors kind of shifted the emphasis from improving spatiotemporal variability of dust to improving dust size distribution. My only concern here is that there is little evaluation on the posterior dust size distribution by comparing simulations against observations, which makes this paper's ending not very strong. At current stage I think authors could argue DA changes size distribution, but it is less obvious that DA improves it. Of course the posterior error decreases, but it does not necessarily mean improvements in the simulations.

I think there are little in-situ size distribution data from field campaigns, so there is not much to do about it. But, I would just suggest that authors make a simple plot on the prior and posterior dust volume size distributions from WRF (e.g., Ryder et al., 2018, Figs. 5-7; Li et al. 2022, Fig. 6), for grids containing the AERONET site and the two deserts. This will be helpful if future in-situ measurements come up from Chinese field campaigns. This also helps readers more directly visualize how DA changes the prior WRF dust size distribution to the posterior. (Fig. S3 is also very helpful by breaking things down into bins.)

Optional: Authors could also do a simple comparison against AERONET inversion data for some references of size distribution (e.g., https://aeronet.gsfc.nasa.gov/cgi-bin/data_display_inv_v3?site=Dalanzadgad&nachal=0&year=2021&month=3&day=16&aero_water=0&level=2&if_day=0&if_err=0&place_code=10&DATA_TYPE=76&year_or_month=0). Doing a small cross-validation for the AERONET size distribution and the WRF size distribution will make the paper's position stronger. I think this suggestion is more optional since AERONET inversion data also has uncertainties based on inversion of remote sensing data.

Other comments:

Figure 5: I am interested in Fig. 5 in terms of why there are increases over the Taklimakan and some regions of the Gobi but decreases over the majority of the Gobi. I think a science explanation is needed in the main text, but I could not find it.

Lines 247-250: I think there needs a science explanation on why or how adding AE changes dust emissions more so in bin 1 and bin 3 than other.

Figs. 7-8: Please add site labels to figure caption so people don't need to scroll back to Fig. 1 to look for what the colors mean.

Fig. 3a-d: Please indicate the unit of dust emissions on the colorbars or in the figure caption. Same for Fig. 5.

In Fig. 1a: Please indicate what the color scale means (although we somehow can guess it).

References:

Ryder, C. L., Marenco, F., Brooke, J. K., Estelles, V., Cotton, R., Formenti, P., McQuaid, J. B., Price, H. C., Liu, D., Ausset, P., Rosenberg, P. D., Taylor, J. W., Choularton, T., Bower, K., Coe, H., Gallagher, M., Crosier, J., Lloyd, G., Highwood, E. J., and Murray, B. J.: Coarse-mode mineral dust size distributions, composition and optical properties from AER-D aircraft measurements over the tropical eastern Atlantic, Atmos. Chem. Phys., 18, 17225–17257, https://doi.org/10.5194/acp-18-17225-2018, 2018.

Li, L., Mahowald, N. M., Kok, J. F., Liu, X., Wu, M., Leung, D. M., Hamilton, D. S., Emmons, L. K., Huang, Y., Sexton, N., Meng, J., and Wan, J.: Importance of different parameterization changes for the updated dust cycle modeling in the Community Atmosphere Model (version 6.1), Geosci. Model Dev., 15, 8181–8219, https://doi.org/10.5194/gmd-15-8181-2022, 2022.

---

## Author Response (AR2)

**Response to the Comments of Referees**

We would like to thank to the reviewers for giving constructive criticisms, which are very helpful in improving the quality of the manuscript. We have made minor revision based on the critical comments and suggestions of the referees. The referee's comments are reproduced (*black*) along with our replies (blue) and changes made to the text (red) in the revised manuscript. All the authors have read the revised manuscript and agreed with submission in its revised form.

**Anonymous Referee #1**

**Comment NO.1:** *Thanks for spending a lot of effort revising the manuscript following the comments and suggestions. I think most of my comments and suggestions are addressed. I just wanted to echo what the 2nd reviewer has said, that the DA gave a relatively homogeneous dust emission change across the domain (Fig. 3a and 3b-d). This makes sense because the AERONET sites are far away from the two deserts. I think partly due to this reason, in the introduction and conclusion sections the authors kind of shifted the emphasis from improving spatiotemporal variability of dust to improving dust size distribution. My only concern here is that there is little evaluation on the posterior dust size distribution by comparing simulations against observations, which makes this paper's ending not very strong. At current stage I think authors could argue DA changes size distribution, but it is less obvious that DA improves it. Of course the posterior error decreases, but it does not necessarily mean improvements in the simulations.*

*I think there are little in-situ size distribution data from field campaigns, so there is not much to do about it. But, I would just suggest that authors make a simple plot on the prior and posterior dust volume size distributions from WRF (e.g., Ryder et al., 2018, Figs. 5-7; Li et al. 2022, Fig. 6), for grids containing the AERONET site and the two deserts. This will be helpful if future in-situ measurements come up from Chinese field campaigns. This also helps readers more directly visualize how DA changes the prior WRF dust size distribution to the posterior. (Fig. S3 is also very helpful by breaking things down into bins.)*

*Optional: Authors could also do a simple comparison against AERONET inversion data for some references of size distribution (e.g., https://aeronet.gsfc.nasa.gov/cgi-bin/data_display_inv_v3?site=Dalanzadgad&nachal=0&year=2021&month=3&day=16& aero_water=0&level=2&if_day=0&if_err=0&place_code=10&DATA_TYPE=76&year_or_m onth=0). Doing a small cross-validation for the AERONET size distribution and the WRF size distribution will make the paper's position stronger. I think this suggestion is more optional since AERONET inversion data also has uncertainties based on inversion of remote sensing data.*

*References:*

*Ryder, C. L., Marenco, F., Brooke, J. K., Estelles, V., Cotton, R., Formenti, P., McQuaid, J. B., Price, H. C., Liu, D., Ausset, P., Rosenberg, P. D., Taylor, J. W., Choularton, T., Bower, K., Coe, H., Gallagher, M., Crosier, J., Lloyd, G., Highwood, E. J., and Murray, B. J.: Coarse-mode mineral dust size distributions, composition and optical properties from AER-D aircraft measurements over*

*the tropical eastern Atlantic, Atmos. Chem. Phys., 18, 17225–17257, https://doi.org/10.5194/acp-18-17225-2018, 2018.*

*Li, L., Mahowald, N. M., Kok, J. F., Liu, X., Wu, M., Leung, D. M., Hamilton, D. S., Emmons, L. K., Huang, Y., Sexton, N., Meng, J., and Wan, J.: Importance of di2erent parameterization changes for the updated dust cycle modeling in the Community Atmosphere Model (version 6.1), Geosci. Model Dev., 15, 8181–8219, https://doi.org/10.5194/gmd-15-8181-2022, 2022.*

**Response:** Done. The normalized atmospheric volume distribution in Gobi desert, Taklimakan desert, and downwind area including AERONET sites for the four experiments during 14-17 March 2021 and 18-23 March 2021 is given in Fig. S6.

**Changes in Manuscript:** Please refer the caption of Fig. S6 in the Supplements.

**Comment NO.2:** *Figure 5: I am interested in Fig. 5 in terms of why there are increases over the Taklimakan and some regions of the Gobi but decreases over the majority of the Gobi. I think a science explanation is needed in the main text, but I could not find it.*

**Response:** Done. Compared with the FR experiment, the dust emission in the AOT+AE DA-SZD experiment is increased over the Taklimakan desert and some regions of the Gobi desert but decreased over the majority of the Gobi desert. This is because the dust emission periods vary across different grids, leading to the opposite trends among different dust source regions.

**Changes in Manuscript:** Please refer the description in the revised manuscript, Page 11 Line 279-282.

**Comment NO.3:** *Lines 247-250: I think there needs a science explanation on why or how adding AE changes dust emissions more so in bin 1 and bin 3 than other.*

**Response:** Done. The significant decrease in bin 1 is due to bin 1 having the highest dust extinction efficiency, and the significant increase in bin 3 is due to bin 3 having the largest proportion in the fine-mode dust emission.

**Changes in Manuscript:** Please refer the description in the revised manuscript, Page 10 Line 256-257.

**Comment NO.4:** *Figs. 7-8: Please add site labels to figure caption so people don't need to scroll back to Fig. 1 to look for what the colors mean.*

**Response:** Done. The site labels are added to figure caption.

**Changes in Manuscript:** Please refer the caption of Figs. 7-8 in the revised manuscript.

**Comment NO.5:** *Fig. 3a-d: Please indicate the unit of dust emissions on the colorbars or in the figure caption. Same for Fig. 5.*

**Response:** Done. The unit of dust emissions is given in caption of Fig. 3 and Fig. 5.

**Changes in Manuscript:** Please refer the caption of Fig. 3 and Fig. 5 in the revised manuscript.

**Comment NO.6:** *In Fig. 1a: Please indicate what the color scale means (although we somehow can guess it).*

**Response:** Done. The meaning of color scale is given in the figure caption.

**Changes in Manuscript:** Please refer the caption of Fig. 1 in the revised manuscript.

**Anonymous Referee #2**

**Comment NO.1:** *The study investigates the effect of assimilating Aerosol Optical Depth (AOD) at 550nm and Angstrom Exponent (AE) between 440 to 870nm simultaneously for a dust event over East Asia in 2022. The authors attempt to explain the added value of assimilating AOD+AE compared to only AOD by conducting assimilation experiments with WRF-Chem where the emissions of each bin are perturbed independently. In addition, a distinctive experiment where the emissions of each bin are perturbed with the same lognormal distribution is conducted to highlight the limitation of that set up. The evaluation is done with both assimilated (AERONET) and independent (SONET and CALIOP) observations, which further solidifies the validity of the results. Overall, the figures are well presented and discussed. As in prior studies, the use of additional optical properties (except AOD) is highlighted, hence I believe the topic is relevant to ACP scope. Since this is a resubmission, I had a brief look at the initial version and confirmed that a lot has been improved (period expanded, statistics, text). I recommend the publication of the manuscript after addressing the following issues:*

*The drawback of the current version is the language and the grammatical mistakes on parts of the manuscript, which sometimes get in the way of understanding parts of the study (see specific comments and technical corrections). I am sure that this can be much improved by the author if they do a thorough re-check of all the text.*

**Response:** We thank the referee for this very positive assessment of our manuscript. The thorough re-check of all the text has been done.

**Comment NO.2:** *In addition, defining a model uncertainty based on either prior studies or on some kind of evaluation analysis is essential. At the moment a specific standard deviation (0.6) is set for the perturbation lognormal distribution. If this is set higher than the true model uncertainty the assimilation will falsely trust the observations more and if this is set lower than the true model uncertainty the assimilation will falsely trust the model more. I would highly recommend being more specific on that matter.*

**Response:** Refer to Dai et al. (2019), the standard deviation of 0.6 corresponds to the uncertainty of the dust emissions for 14 global models (Huneeus et al., 2011).

References:

Dai, Cheng, Goto, Schutgens, Kikuchi, Yoshida, et al. Inverting the East Asian Dust Emission Fluxes Using the Ensemble Kalman Smoother and Himawari-8 AODs: A Case Study with WRF-Chem v3.5.1. Atmosphere, 10(9), 543. https://doi.org/10.3390/atmos10090543, 2019.

Huneeus, N., Schulz, M., Balkanski, Y., Griesfeller, J., Prospero, J., Kinne, S., Bauer, S., Boucher, O., Chin, M., Dentener, F., Diehl, T., Easter, R., Fillmore, D., Ghan, S., Ginoux, P., Grini, A., Horowitz, L., Koch, D., Krol, M. C., Landing, W., Liu, X., Mahowald, N., Miller, R., Morcrette, J.-J., Myhre, G., Penner, J., Perlwitz, J., Stier, P., Takemura, T., and Zender, C. S.: Global dust model intercomparison in AeroCom phase I, Atmospheric Chemistry and Physics, 11, 7781–7816, https://doi.org/10.5194/acp-11-7781-2011, 2011.

**Changes in Manuscript:** Please refer the description in the revised manuscript, Page 7 Line 206-207.

**Comment NO.3:** *L17-19: This sentence needs to be rephrased.*

**Response:** Accept. The sentence has been modified as "Due to the lack of observation during dust storms and the accuracy of satellite-retrieved AE depending on the instrument and retrieval algorithm, it is possible to estimate the dust storm emission using the time-lagged ground-based AE observations."

**Changes in Manuscript:** Please refer the description in the revised manuscript, Page 1 Line 17-19.

**Comment NO.4:** *L17: What do you mean by "lagged" ground observations? Lagged in the and space, as the dust was emitted a few days earlier in the source region before it was transported over the observation site? Could you be more specific and rephrase?*

**Response:** The "lagged" means the observation at a later time contains the information about the dust emitted a few days earlier in the source region. We have replaced "lagged" with "time-lagged".

**Changes in Manuscript:** Please refer the description in the revised manuscript, Page 1 Line 17-19.

**Comment NO.5:** *L24: This sentence needs to be rephrased. In the current form the reader cannot determine where these improvements are referring to. Be more specific on the experiments you are comparing, the reference you are using, which variables you are evaluating and the metrics you calculate. Smaller sentences always help.*

**Response:** Done. The sentence has been rewritten as "Validation by independent observations from Skynet Observation NETwork (SONET) shows that assimilating additional AE information reduces the root mean square error (RMSE) of simulated AOT and AE by approximately 17% and 61% respectively, as shown by the comparison between the AOT DA-SZD and AOT+AE DA-SZD experiments."

**Changes in Manuscript:** Please refer the description in the revised manuscript, Page 1 Line 23-26.

**Comment NO.6:** *L25-26: "can be improved" or "it is improved"?*

**Response:** Agree, "can be improved" has been replaced by "are improved".

**Changes in Manuscript:** Please refer the description in the revised manuscript, Page 1 Line 26-27.

**Comment NO.7:** *L45-47: Consider looking at the AEROCOM I and AEROCOM III studies. The simulated dust size range and model resolution are major factors for the diversity we observe in the simulated total emission flux.*

**Response:** Done. The sentence has been modified as "However, due to differences in the parameterizations of dust source fluxes, dust particle sizes, and model resolutions, simulated East Asian dust emissions vary by more than an order of magnitude among different models (Textor et al., 2006; Uno et al., 2006; Gliß et al., 2021; Kok et al., 2021), indicating that dust emission is a highly uncertain process in dust simulation."

References:

Textor, C., Schulz, M., Guibert, S., Kinne, S., Balkanski, Y., Bauer, S., Berntsen, T., Berglen, T., Boucher, O., Chin, M., Dentener, F., Diehl, T., Easter, R., Feichter, H., Fillmore, D., Ghan, S.,

Ginoux, P., Gong, S., Grini, A., Hendricks, J., Horowitz, L., Huang, P., Isaksen, I., Iversen, T., Kloster, S., Koch, D., Kirkeva, A., Kristjansson, J. E., Krol, M., Lauer, A., Lamarque, J. F., Liu, X., Montanaro, V., Myhre, G., Penner, J., Pitari, G., Reddy, S., Seland, Ø., Stier, P., Takemura, T., and Tie, X.: Analysis and quantification of the diversities of aerosol life cycles within AeroCom, Atmos. Chem. Phys., 37, 2006.

Gliß, J., Mortier, A., Schulz, M., Andrews, E., Balkanski, Y., Bauer, S. E., Benedictow, A. M. K., Bian, H., Checa-Garcia, R., Chin, M., Ginoux, P., Griesfeller, J. J., Heckel, A., Kipling, Z., Kirkevåg, A., Kokkola, H., Laj, P., Le Sager, P., Lund, M. T., Lund Myhre, C., Matsui, H., Myhre, G., Neubauer, D., Van Noije, T., North, P., Olivié, D. J. L., Rémy, S., Sogacheva, L., Takemura, T., Tsigaridis, K., and Tsyro, S. G.: AeroCom phase III multi-model evaluation of the aerosol life cycle and optical properties using ground- and space-based remote sensing as well as surface in situ observations, Atmos. Chem. Phys., 21, 87–128, https://doi.org/10.5194/acp-21-87-2021, 2021.

**Changes in Manuscript:** Please refer the description in the revised manuscript, Page 2 Line 46-49.

**Comment NO.8:** *L64-66: Also consider the series of paper by Chen (2018 ACP, 2019 ACP).*

**Response:** Done. The estimated emission may be misrepresented by not including observations related to size (Chen et al., 2018, 2019; Tsikerdekis et al., 2021).

References:

Chen, C., Dubovik, O., Henze, D. K., Lapyonak, T., Chin, M., Ducos, F., Litvinov, P., Huang, X., and Li, L.: Retrieval of desert dust and carbonaceous aerosol emissions over Africa from POLDER/PARASOL products generated by the GRASP algorithm, Atmos. Chem. Phys., 18, 12551–12580, https://doi.org/10.5194/acp-18-12551-2018, 2018.

Chen, C., Dubovik, O., Henze, D. K., Chin, M., Lapyonok, T., Schuster, G. L., Ducos, F., Fuertes, D., Litvinov, P., Li, L., Lopatin, A., Hu, Q., and Torres, B.: Constraining global aerosol emissions using POLDER/PARASOL satellite remote sensing observations, Atmos. Chem. Phys., 19, 14585–14606, https://doi.org/10.5194/acp-19-14585-2019, 2019.

**Changes in Manuscript:** Please refer the description in the revised manuscript, Page 3 Line 67-68.

**Comment NO.9:** *L98-100: Could you provide the instrument error for AE?*

**Response:** Done. The instrument error in AOT ($\epsilon_{AOTi}$) is defined as 0.015 and the instrument error of AE ($\epsilon_{AEi}$) is estimated by propagating the instrument error in AOT at 440 and 870 nm as $\epsilon_{AEi} = \sqrt{((\epsilon_{AOTi}/\tau_{870})^2 + (\epsilon_{AOTi}/\tau_{440})^2)/(ln(870/440)^2)}$ (Schutgens et al., 2010).

**Changes in Manuscript:** Please refer the description in the revised manuscript, Page 4 Line 100-102.

**Comment NO.10:** *L199-201: It is not clear if you are using the same perturbation factor across your whole domain or if you are defining your perturbation per grid. Could you please mention it here? Also please consider revising this sentence, it is very hard to read.*

**Response:** We use the same perturbation factor across the whole domain. This sentence has been modified as "20 ensemble members are generated by perturbing the emission fluxes in each of the five dust bins. The same perturbation factor is used across the whole domain. The random

perturbation factor is drawn from a lognormal distribution with a mean of 1 and a standard deviation of 0.6."

**Changes in Manuscript:** Please refer the description in the revised manuscript, Page 7 Line 204-206.

**Comment NO.11:** *L201: The lognormal distribution spread is set to 0.6. By spread you mean standard deviation? Also, since this value determines the emission/optical property uncertainty of your ensemble, is it based on a specific analysis you conducted or based on a prior study?*

**Response:** Yes, spread means the standard deviation. Refer to Dai et al. (2019), the standard deviation of 0.6 corresponds to the uncertainty of the dust emissions for 14 global models (Huneeus et al., 2011).

References:

Dai, Cheng, Goto, Schutgens, Kikuchi, Yoshida, et al. Inverting the East Asian Dust Emission Fluxes Using the Ensemble Kalman Smoother and Himawari-8 AODs: A Case Study with WRF-Chem v3.5.1. Atmosphere, 10(9), 543. https://doi.org/10.3390/atmos10090543, 2019.

Huneeus, N., Schulz, M., Balkanski, Y., Griesfeller, J., Prospero, J., Kinne, S., Bauer, S., Boucher, O., Chin, M., Dentener, F., Diehl, T., Easter, R., Fillmore, D., Ghan, S., Ginoux, P., Grini, A., Horowitz, L., Koch, D., Krol, M. C., Landing, W., Liu, X., Mahowald, N., Miller, R., Morcrette, J.-J., Myhre, G., Penner, J., Perlwitz, J., Stier, P., Takemura, T., and Zender, C. S.: Global dust model intercomparison in AeroCom phase I, Atmospheric Chemistry and Physics, 11, 7781–7816, https://doi.org/10.5194/acp-11-7781-2011, 2011.

**Changes in Manuscript:** Please refer the description in the revised manuscript, Page 7 Line 206-207.

**Comment NO.12:** *L209: As a general comment here: There is model spin-up as well as data assimilation spin-up (see Tsikerdekis et al., 2022). Are you referring to model spin-up here? Note that emissions may be corrected even in the first cycle (fully in the 6th cycle on your system), but it may take several days to see the effect on the aerosol optical properties, assuming of course you have a continues flow of assimilated observations.*

**Response:** This is not referring to model spin-up and we have modified the statement.

The initial condition at 12:00 UTC on 11 March 2021 is prepared by an 11-day simulation executed by WRF-Chem without any aerosol data assimilation as a spin-up. The results from 12:00 UTC on 11 March 2021 to 23:59 UTC on 13 March 2021 are excluded in the analysis.

**Changes in Manuscript:** Please refer the description in the revised manuscript, Page 7 Line 198-200 and Page 8 Line 214-215.

**Comment NO.13:** *L209-210: Is the baseline experiment an ensemble run or a single run? The difference between the two may be minuscule since you lognormal distribution has a mean of 1, but still it is important to check the differences per bin. If it is an ensemble run, which perturbation set-up is using, the DA-SZD or the regular SZD?*

**Response:** The baseline experiment is a single run. We have checked that the differences per bin are limited.

**Changes in Manuscript:** Please refer the description in the revised manuscript, Page 8 Line 215.

**Comment NO.14:** *L84: "in global" to"globally"*

**Response:** Done. "in global" has been replaced by "globally".

**Changes in Manuscript:** Please refer the description in the revised manuscript, Page 3 Line 86.

**Comment NO.15:** *L88: "of the ones at" to "of the AOTs at"*

**Response:** Done. "of the ones at" has been replaced by "of the AOTs at".

**Changes in Manuscript:** Please refer the description in the revised manuscript, Page 3 Line 90.

**Comment NO.16:** *L150: "as same as" to "the in"*

**Response:** Done. "as same as" has been replaced by "the in".

**Changes in Manuscript:** Please refer the description in the revised manuscript, Page 6 Line 153.

**Comment NO.17:** *L196: "has uncertainties" to "uncertainties"*

**Response:** Done. "has uncertainties" has been replaced by "uncertainties".

**Changes in Manuscript:** Please refer the description in the revised manuscript, Page 7 Line 200.

**Comment NO.18:** *L197: "in simulation" can be deleted.*

**Response:** Done. "in simulation" has been deleted.

**Changes in Manuscript:** Please refer the description in the revised manuscript, Page 7 Line 201.

**Comment NO.19:** *L198: "only assimilates" to "assimilated only"*

**Response:** Done. "only assimilates" has been replaced by "assimilated only".

**Changes in Manuscript:** Please refer the description in the revised manuscript, Page 7 Line 202.

**Comment NO.20:** *L206: "as same as" to "as"*

**Response:** Done. "as same as" has been replaced by "as".

**Changes in Manuscript:** Please refer the description in the revised manuscript, Page 8 Line 213.

**Comment NO.21:** *L207: "except the" to "except that the"*

**Response:** Done. "except the" has been replaced by "except that the".

**Changes in Manuscript:** Please refer the description in the revised manuscript, Page 8 Line 213.

**Comment NO.22:** *L207: "with same" to "with the same"*

**Response:** Done. "with same" has been replaced by "with the same".

**Changes in Manuscript:** Please refer the description in the revised manuscript, Page 8 Line 214.

**Comment NO.23:** *L209: "as the spin-up" to "as spin-up"*

**Response:** Done. "as the spin-up" has been deleted.

**Changes in Manuscript:** Please refer the description in the revised manuscript, Page 8 Line 215.

**Comment NO.24:** *L274: "with additional AE" to "with the addition of AE"*

**Response:** Done. "with additional AE" has been replaced by "with the addition of AE".

**Changes in Manuscript:** Please refer the description in the revised manuscript, Page 12 Line 286.

**Comment NO.25:** *L301: "due to the uncertainty" to "since the uncertainty"*

**Response:** Done. "due to the uncertainty" has been replaced by "since the uncertainty".

**Changes in Manuscript:** Please refer the description in the revised manuscript, Page 14 Line 314.

**Comment NO.26:** *L302: "Those" to "These results"*

**Response:** Done. "Those" has been replaced by "These results".

**Changes in Manuscript:** Please refer the description in the revised manuscript, Page 14 Line 315.

**Comment NO.27:** *L303: "to the optimization" to "for the optimization"*

**Response:** Done. "to the optimization" has been replaced by "for the optimization".

**Changes in Manuscript:** Please refer the description in the revised manuscript, Page 14 Line 316.

**Comment NO.28:** *L303: "better" to "the"*

**Response:** Done. "better" has been replaced by "the".

**Changes in Manuscript:** Please refer the description in the revised manuscript, Page 14 Line 316.

**Comment NO.29:** *L304: "The similar" to "Similar"*

**Response:** Done. "The similar" has been replaced by "Similar".

**Changes in Manuscript:** Please refer the description in the revised manuscript, Page 14 Line 317.

**Comment NO.30:** *L304: "with assimilated" to "with the assimilated"*

**Response:** Done. "with assimilated" has been replaced by "with the assimilated".

**Changes in Manuscript:** Please refer the description in the revised manuscript, Page 14 Line 317.